# Fat Quality Impacts the Effect of a High-Fat Diet on the Fatty Acid Profile, Life History Traits and Gene Expression in *Drosophila melanogaster*

**DOI:** 10.3390/cells11244043

**Published:** 2022-12-14

**Authors:** Virginia Eickelberg, Gerald Rimbach, Yvonne Seidler, Mario Hasler, Stefanie Staats, Kai Lüersen

**Affiliations:** 1Institute of Human Nutrition and Food Science, University of Kiel, Hermann-Rodewald-Str. 6, D-24118 Kiel, Germany; 2Lehrfach Variationsstatistik, University of Kiel, Hermann-Rodewald-Str. 9, D-24118 Kiel, Germany

**Keywords:** *Drosophila melanogaster*, high-fat diet, fat quantity, fat quality, gene expression

## Abstract

Feeding a high-fat diet (HFD) has been shown to alter phenotypic and metabolic parameters in *Drosophila melanogaster*. However, the impact of fat quantity and quality remains uncertain. We first used butterfat (BF) as an example to investigate the effects of increasing dietary fat content (3–12%) on male and female fruit flies. Although body weight and body composition were not altered by any BF concentration, health parameters, such as lifespan, fecundity and larval development, were negatively affected in a dose-dependent manner. When fruit flies were fed various 12% HFDs (BF, sunflower oil, olive oil, linseed oil, fish oil), their fatty acid profiles shifted according to the dietary fat qualities. Moreover, fat quality was found to determine the effect size of the response to an HFD for traits, such as lifespan, climbing activity, or fertility. Consistently, we also found a highly fat quality-specific transcriptional response to three exemplary HFD qualities with a small overlap of only 30 differentially expressed genes associated with the immune/stress response and fatty acid metabolism. In conclusion, our data indicate that not only the fat content but also the fat quality is a crucial factor in terms of life-history traits when applying an HFD in *D. melanogaster*.

## 1. Introduction

Over the last few decades, the eating behavior of humans in Western societies has been characterized by an increased consumption of high-energy foods comprising high fat, sugar and salt levels but only a low nutrient content, best known as the “Westernization” of the food pattern. High energy intake combined with insufficient physical activity promote the development of obesity and type-2 diabetes, which leads to remarkable global public-health challenges [1,2,3]. The global supply and consumption of dietary fats and oils from animal- and plant-derived sources is continuously increasing, whereas the majority of dietary fats are provided by vegetable fats and oils (e.g., derived from palm, soybean, sunflower and rapeseed). These nutritional fats and oils often differ considerably in their fatty acid composition [4]. Fat quality has been recognized as an important factor that determines the nutritional value of oils and fats owing to its effect on the lipid metabolism and health parameters [5]. Accordingly, global dietary recommendations for humans consider a reduction of total fat (<30% of energy), saturated fat (<10% of energy) and *trans*-fat intake (<1% of energy) by replacement with unsaturated (especially polyunsaturated) fats for a healthy diet [6,7]. The exchange of saturated fatty acids (SFAs) with mono- and polyunsaturated fatty acids (MUFAs, PUFAs) is usually recommended to reduce chronic diseases related to the so-called metabolic syndrome. However, in particular, the impact of total fat and saturated fat in the diet has been the object of debate [4] and inconsistent findings were reported for total fat intake and the risk of chronic diseases, such as obesity [8], diabetes [9] and cardiovascular diseases [10] in humans.

By applying the model organism *D. melanogaster*, the short- and long-term effects of feeding a high-energy diet can be systematically examined [11,12]. The fruit fly is characterized by a high reproductive rate, a short life cycle and a relatively short lifespan of 60–80 days. Hence, the impact of dietary interventions on life-history traits, such as feeding, locomotor behavior, longevity, metabolism, reproductive capacity, and stress tolerance can be studied within a relatively short period of time and cost-effectively since sophisticated experimental equipment is often not required [12]. In addition, most of the core metabolic and signal transduction pathways are evolutionary highly conserved between *D. melanogaster* and mammals [13]. 

*D. melanogaster* originates from western tropical habitats in Africa and can be categorized as omnivore [14], that preferably subsists of sugar-rich fruits and fermented plant-based products, and alternatively occupy animal-based products (e.g., cadavers, excrements) serving as a breeding site [15]. Both naturally occurring food sources and artificial laboratory diets of fruit flies are usually rich in carbohydrates and proteins, and relatively low in fat. A common fat source in experimental diets is inactive yeast that provides a fat content of approximately 5%. Nevertheless, fruit flies hold an enzymatic setup that apart from proteases and glycosylases also features lipase cleaving abilities (the *Drosophila* genome encodes several putative lipases exhibiting midgut expression) allowing for the utilization of dietary fats [16,17].

Feeding energy-rich food to fruit flies can be accomplished via high-sugar diets (HSD) and high-fat diets (HFD). For an HFD, the rather low-fat *D. melanogaster* standard food medium has been usually enriched with the fat sources coconut fat or lard to a final concentration of 20–30% and 15% (*w*/*v*), respectively [12]. Importantly, the terms “HSD” and “HFD” are not strictly defined, leading to variations in the applied diets and feeding regimes across studies [18]. 

Previous investigations revealed that ingestion of an HFD adversely affects the climbing ability [19,20,21,22,23], mating behavior [24], fecundity [20], and temperature tolerance (e.g., cold stress) in fruit flies [20,25]. A short-term feeding of an HFD with 5–30% (*w*/*v*) coconut fat caused phenotypic and metabolic changes in *D. melanogaster* manifested in obesity and related comorbidities, such as dysfunction of the heart [19,21,26] and the muscles [23] as well as in an alteration of the metabolic profile including an increase in whole-body triacylglycerol (TAG) and glucose levels and changes in insulin signaling [21,23,25]. Moreover, feeding an HFD with coconut fat [19,20,23,25], lard [27,28] or palmitic acid [29] throughout life resulted in a reduction of lifespan in *D. melanogaster*.

The present study was conducted to elucidate the impact of quantity and quality of dietary fats on health parameters in the model organism *D. melanogaster*. Firstly, HFD with increasing concentrations of BF were fed to male and female fruit flies and the short-term impact on food intake, body weight, body composition and reproduction was examined. Moreover, the influence of long-term feeding of different BF quantities on the lifespan of *D. melanogaster* was investigated. BF as a high source of SFAs [30] was used instead of coconut fat, as physical properties of coconut fat (e.g., melting point, which makes the flies stick to the medium and might falsify survivor curves [31]) partly limit its applicability in *D. melanogaster* HFD long-term studies, such as lifespan experiments. Secondly, HFD based on different fat sources were used to evaluate the impact of fat quality on body composition, several life-history traits and differential gene expression of *D. melanogaster*. Adapted to the global shift in choice of dietary fats from animal- to plant-based sources in recent years [4], HFDs were prepared using also the plant-derived oils sunflower oil (SO), olive oil (OO) and linseed oil (LO) in our *Drosophila* feeding trials. Moreover, the oils were selected based on a diverse range of fatty acid compositions. Besides BF with mainly SFAs [30], FO exhibits nearly equal proportions of SFAs, MUFAs and PUFAs, while OO predominantly contains MUFAs and both SO and LO are mainly composed of PUFAs. Our studies revealed that both fat quantity and fat quality are important factors that must be considered when performing HFD feeding trials with *D. melanogaster*.

## 2. Materials and Methods

### 2.1. Fly Strains and Husbandry

*D. melanogaster* strain *w^1118^* (Bloomington Drosophila Stock Center #5905, Indiana University, Bloomington, IN, USA) was used for all experiments. Flies were maintained on modified Caltech (CT) medium (6.0% dextrose, 3.0% sucrose (Carl Roth, Karlsruhe, Germany), 6.0% cornmeal (Kisker, Steinfurt, Germany), 3.0% inactive dry yeast (contributing 1.6% protein, 1% complex carbohydrates and 0.15% fat as well as vitamins and minerals according to the product information), 1.0% agar, 1.0% nipagin (Genesee Scientific, San Diego, CA, USA) (dissolved in ethanol (20% *w*/*v*) (VWR, Radnor, PA, USA) and 0.3% propionic acid (Carl Roth, Karlsruhe, Germany)) in *Drosophila* storage bottles (Kisker, Steinfurt, Germany) at 25 °C and 60% relative humidity with a 12/12 h light/dark cycle in a climate chamber (Memmert, Schwabach, Germany) [32]. 

### 2.2. Synchronization of Eggs

For collection of synchronized eggs, 150 to 200 adult flies were anaesthetized and placed in an egg-laying cage with a grape agar plate spread with a thin layer of active yeast (all Kisker, Steinfurt, Germany) dissolved in water. The next day, the grape agar plate was replaced ones. After additional 24 h of oviposition, the adult flies were discarded. The surface of the grape agar plate was washed with PBS (PAN-Biotech, Aidenbach, Germany) and the eggs were transferred to a 15 mL tube. The egg/PBS suspension was rinsed three times with PBS and 32 µL of the eggs were distributed into *Drosophila* bottles with CT medium. The bottles were maintained in an incubator under standard conditions. After eclosion, the flies were transferred to standard SY food for mating. After two days, the age-matched flies were used for further experiments [33]. 

### 2.3. Preparation of High-Fat Diets

The different experimental HFDs were based on a standard sugar yeast (SY) medium (control medium) containing 10% sucrose, 10% inactive dry yeast (contributing 5.2% protein, 3.3% complex carbohydrates and 0.5% fat of which 0.1–0.2% are SFAs and 0.2–0.4% MUFAs, as well as vitamins and minerals; product information, Genesee Scientific), 2.0% agar, 1.5% nipagin (dissolved in ethanol (20% *w*/*v*) and 0.3% propionic acid. For the evaluation of the influence of fat quantity, 3% (HFD-3), 6% (HFD-6), 9% (HFD-9) or 12% BF (HFD-12; all (*w*/*v*)) was added to the SY medium. Fat quality studies were carried out with control medium that was supplemented with 12% (*w*/*v*) BF (HFD-BF), SO (HFD-SO), OO (HFD-OO), LO (HFD-LO) and FO (HFD-FO), respectively. If not otherwise specified, the *Drosophila* vials (Kisker, Steinfurt, Germany) were placed horizontally for maintenance under standard conditions.

### 2.4. Determination of the Fatty Acid Profile of Fruit Flies and Supplemented Oils and Fats

To determine the fatty acid profile of *D. melanogaster*, 725 male and 465 female flies, respectively, were pre-fed in *Drosophila* bottles with control or HFD under standard conditions for 3 days. After a short starvation period of 1 h, the flies were transferred to a 15 mL tube and weighed with a precision scale (Sartorius, Goettingen, Germany). The flies were grinded in 3 mL PBS in 15 mL tubes using an ultrasonic homogenizer model FB 120 (Thermo Fisher Scientific, Darmstadt, Germany) and a disperser model T10 (IKA, Staufen, Germany). The analysis of the fatty acid profile of flies and oils was conducted with the preparation of fatty acid methyl esters (FAME) using trimethylsulfonium hydroxide according to ASU L 13.00-27/3:2018-06 and DIN EN ISO 12966-3:2016-11. FAME were analyzed by capillary gas chromatography (GC) using a Shimadzu GC-2010 Plus GC system equipped with a split/splitless injector AOC-20i, an automated liquid sampler AOC-20s and a flame ionization detector (Shimatsu GmbH, Duisburg, Germany). The inlet was operated in constant flow mode with helium carrier gas and a 100:1 split ratio. 1 µL of each sample was separated on a capillary GC column (Zebron^®^ ZB-FAME, 0.20 µm, 30 m × 0.25 mm ID; Phenomenex Ltd., Aschaffenburg, Germany) at a flow rate of 1.2 mL/min with a total run time of 30.33 min employing a temperature program starting with an initial temperature of 80 °C for 2 min, heated by a rate of 15 °C/min to 140 °C, then elevated by a heat rate of 2.5 °C/min to 190 °C, and finally heated by 30 °C/min to 260 °C, before the temperature was hold for 2 min). FAMEs were analyzed with the flame ionization detector according to ASU L 13.00-45/46:2018-06 and DIN EN ISO 12966-1/4:2015-11 [SGS Analytics Germany GmbH, Jena]. Peak identification was carried out via external standards (Merck-Supelco, Darmstadt, Germany). Data were calculated as % identified FAME.

### 2.5. Determination of Food Intake

The measurement of food intake was based on the Excreta quantification (EX-Q)-method [34]. In brief, 3-day-old flies were either directly used in the EX-Q assay or first pre-fed for additional 7 days on control and HFD, respectively. The food intake assay was started by placing 25 flies into an empty vial with a small feeding cap (diameter: 25 mm, lids of small Eppendorf tubes) containing 1% (*w*/*v*) Brilliant Blue FCF (Omikron, Rietberg, Germany) stained experimental food medium. The blue dye cannot penetrate intact *Drosophila* intestinal epithelium. Flies were allowed to ingest food for 24 h, while the blue feeding cap was replaced once. Afterwards, the flies were removed and the excretions (fecal spots) in the vial were rinsed with distilled water. Absorption was determined with an iEMS 96 well Microplate Reader (Labsystems Oy, Helsinki, Finland) at a wavelength of 620 nm. The ingested food quantity and its energy content was calculated per fly and per day. The experiments were conducted three times in triplicate.

### 2.6. Determination of Body Weight and Body Composition

Twenty age- and sex-matched flies per vial were pre-fed with a SY control or HFD under standard conditions for 5 days. After a 1 h starvation period, the flies of each group were pooled and weighed with a precision scale. The average body weight per fly was calculated and the flies were stored at −80 °C for further analysis. 

For determination of whole-body composition, three flies per sample were homogenized in PBS with 0.05% Triton X100 (PBT) (Carl Roth, Karlsruhe, Germany) in a TissueLyser (Qiagen, Hilden, Germany). After centrifugation at 16,000× *g* for 20 min at 4 °C, the supernatants were processed for whole body protein (Pierce BCA Protein Assay Kit), TAG (Infinity kit (both Thermo Fisher Scientific, Darmstadt, Germany)) and glucose (Fluitest GLU (Analyticon, Lichtenfels, Germany)) analyses. All assays were conducted according to the manufacturer’s instructions. Experiments were independently repeated three times in duplicate.

### 2.7. Lifespan Experiments and Determination of Gut Integrity

Three-day-old sex-matched flies were allocated to control or HFD in groups of 25 animal per vial. Flies were maintained under standard conditions. Food medium was renewed 3 times a week while dead flies were counted over time. A total of 2 (fat quantity) and 3 (fat quality) independent lifespan experiments with 100–150 animals per condition were performed, respectively. Gut integrity was monitored over a time period of 30 days following the protocol of Rera et al. [35]. To this end, 3-day-old flies were allocated in groups of 25 animals to *Drosophila* vials containing 1% (*w*/*v*) Brilliant Blue FCF-stained SY and 12% HFD-BF, respectively. The medium was changed 3 times per week and the number of flies that exhibited the smurf phenotype, i.e., the usually non-absorbable dye had entered the fly body owing to a leaky gut epithelium, was recorded.

### 2.8. Induced Climbing Activity Assay

The physical fitness was assessed by measurement of the induced climbing activity according to [36] with modifications. The assay was conducted with 25 flies per vial pre-fed with control or HFD for 3 days. Flies were transferred into 195 mm glass cylinders and allowed to adapt for 2 min, before they were collected on the bottom by tapping the glass cylinder on the table top. The induced movement (negative geotaxis) of the flies was recorded with a video camera (Panasonic, Kadoma, Japan) until all flies had crossed the target line (110 mm), or, alternatively, the experimental time of 1 min run out. The videos were analyzed in two second intervals to determine the cumulative number of flies crossing the target line. The experiment was independently conducted three times in triplicate.

### 2.9. Determination of Spontaneous Locomotor Activity

The spontaneous locomotor activity was recorded using a *Drosophila* Activity Monitoring (DAM) System (TriKinetics, Waltham, MA, USA) as described previously [37]. A total of 20 3-day-old flies per vial were maintained on control or HFD for 5 days (fat quantity) and 3 days (fat quality), respectively, before they were transferred to glass tubes containing the same food as for pre-feeding. The glass tubes were placed horizontally in the DAM System that was integrated in a climate chamber. Following an adaptation period of 24 h, the spontaneous activity of the flies was recorded with the DAM System Collection software (TriKinetics) for 48 h. Data were converted with DAM Filescan110X (TriKinetics) and the spontaneous activity of the flies was gathered at ten-minute intervals. Corresponding actograms were set up and the mean locomotor activity per fly of each group was calculated. All experiments were performed in triplicate.

### 2.10. Determination of Female Fecundity

Synchronized flies were allowed to mate for 2 days. Females were placed onto the experimental diets (2 flies per vial) and transferred to new vials with fresh food every 2–3 days. Laid eggs were counted and the oviposition was monitored for 15 days in total [38]. Fecundity experiments were independently performed three times in triplicate.

### 2.11. Egg-Laying Preference Assay

To examine whether fat supplementation affects the choice of the oviposition site of fruit flies, a two-choice assay was performed [39]. To this end, 25 females per vial in duplicate were pre-fed a control diet under standard conditions for 7 days. Flies were transferred into a glass box with a clip cover, which contained two petri dishes with different combinations of control and experimental diets. The glass boxes were randomly placed in a dark incubator at 25 °C and 60% relative humidity for 24 h. Then, the flies were removed, and the eggs laid per petri dish were counted under a S6 E microscope (Leica, Wetzlar, Germany). The oviposition preference index (PI) was calculated as: PI = (egg number HFD—egg number control food)/(total egg number per box). The PI per duplicate were averaged revealing the preferred food. The two-choice assay was independently conducted three times in duplicate. 

### 2.12. Determination of the Development Rate

A total of 20 synchronized eggs per vial were maintained on experimental diets under standard conditions and the numbers of developed pupae and adult flies were recorded daily [33]. The average percentage of developed pupae and flies over time was calculated. The number of pupae and hatched flies per vial were counted for 12 days and the size of pupae were measured by area. The development rate was examined in three independent experiments in triplicate.

### 2.13. Determination of the Abdominal Width and Ovarian Area

For measurement of the abdominal width and ovarian area, 20 females per vial in triplicate were pre-fed with control or HFD-BF for 15 days. One part of the flies were anaesthetized with triethylamine (Merck, Darmstadt, Germany) for harvesting and were mounted front side upwards with liquid glue on a microscope slide (Carl Roth, Karlsruhe, Germany) [40]. Images were taken under a microscope with an ocular camera AM7025X (Dino-Lite, Almere, The Netherlands). Photos were assessed with the software ImageJ, version 1.52a (Wayne Rasband, Bethesda, MD, USA). The abdominal width was examined in 2 independent experiments with 15 females per treatment. The other part of the flies were shortly anaesthetized with CO_2_ and ovaries were dissected with two forceps [41]. Ovaries were imaged on a microscope slide in PBS with an ocular camera DFC3000 G (Leica, Wetzlar, Germany). The ovarian area was measured in 2 independent experiments with 80 females per treatment using ImageJ software.

### 2.14. Isolation and Sequencing of RNA

RNA isolation and sequencing was carried out following the protocol described in [42]. Briefly, female flies pre-fed with control, HFD-BF, HFD-SO or HFD-FO for three days were homogenized in peqGOLD TriFast^TM^ (VWR, Radnor, USA) using a TissueLyser at 25 Hz for 10 min. RNA isolation was carried out according to manufacturer’s instructions. Purity and quantity of the isolated RNA was assessed spectrophotometrically (NanoDrop™ 2000; Thermo Fisher Scientific, Darmstadt, Germany). RNA samples adjusted to 100 ng/mL with DEPC-treated water were used to prepare libraries for RNA sequencing. It is followed by sequencing and cluster generation employing a HiSeq 3000/HiSeq 4000 system (Illumina, San Diego, CA, USA) and trimming of the sequencing of RNA reads for quality control and removal of adaptors, conducted at the Institute of Clinical Molecular Biology, University of Kiel. The software CLC genomics workbench, version 9.5.2 (QIAGEN, Venlo, The Netherlands/Hilden, Germany) was used for alignment of reads to the *D. melanogaster* genome, which were utilized to initiate read counts, and for evaluation of the differential expression. The expression was considered statistically significant, when *p*-value ≤ 0.05. Experiments were independently repeated three times in duplicate. Separate lists were generated for genes that were significantly upregulated by HFD-BF, HFD-SO and HFD-FO, respectively, when compared to controls consisting of transcripts with an FDR adjusted *p*-value ≤ 0.1 and a fold change > 1.5. Similar lists were created for genes that fulfil the criteria *p*-value < 0.05 and fold change > 1.5 in two (HFD-BF and HFD-FO; HFD-BF and HFD-SO; HFD-SO and HFD-FO) or in all HFD groups. The same was done for downregulated genes with a threshold for the fold change of >−1.5. These gene lists were separately applied to the Database for Annotation, Visualization, and Integrated Discovery (DAVID) bioinformatics software (https://david.ncifcrf.gov/ (accessed on 10 November 2022)) using flybase gene ID as the identifier to unravel significantly affected functional clusters. The annotation of genes and their functions were supported by additional literature and flybase search (https://flybase.org/ (accessed on 10 November 2022)). 

### 2.15. Statistical Analysis

Statistical analyses were carried out using the statistical software R, version 4.0 (R Foundation for Statistical Computing, Vienna, Austria), except for the evaluation of lifespan and analysis of transcriptional data. The data evaluation was initiated with the definition of appropriate statistical mixed models [43] for the measurement variables food intake, energy intake, body weight, TAG, protein, glucose content, climbing success probability, average activity, number of eggs, oviposition preference index, development time and number of pupae/hatched flies from eggs. When both genders were analyzed, the models included fat quantity or fat quality and gender, as well as their interaction term as fixed factors and repetition as a random factor. For the investigation of fecundity, oviposition and development, only fat was defined as a fixed factor and repetition as a random factor. For all measurement variables, except for climbing success probability and fecundity, the residuals were assumed to be normally distributed and to be heteroscedastic with respect to the different levels and types of fat. For fecundity, the residuals were assumed to be homoscedastic. These assumptions are based on graphical residual analyses. For the climbing success probability, the residuals follow a binomial distribution. Based on these models, Pseudo R^2^ were calculated [44] and analyses of variances (ANOVA) were conducted, followed by multiple contrast tests [45] in order to compare the several fat levels or fat types with the control (fat = 0%). For the climbing assay, values are expressed as the success probability per fly in a group to cross the target line within 60 s. This success probability is calculated by conversion of logit values from the above generalized mixed model, including all values obtained in triplicate.

For the comparison of the survival curves, the software GraphPad Prism, version 9.0 (GraphPad Software, San Diego, CA, USA) was used. The survival curves of the HFD groups were compared with the respective control group (fat = 0%) using a logrank (Mantel-Cox) test. 

The Wald test was applied for the statistical analysis of the transcriptional data (CLC Genomics Workbench software (version 9.5.2), Qiagen). The thresholds that define a transcript as differentially expressed were set at a minimum 1.5-fold change and a *p*-value < 0.05.

## 3. Results

### 3.1. Increasing Concentrations of Dietary Butterfat Significantly Increased the Energy Intake without Affecting the Body Composition of Male and Female D. melanogaster

We first examined whether increasing the concentration of BF in a *D. melanogaster* standard SY diet affects the food intake of fruit flies. When tested after 7 days of pre-feeding on the experimental HFD diets, a significant increase in food intake was solely found in males that were fed the diet with the highest BF content (HFD-12). In all other treatment groups, food intake was not affected by BF supplementation when compared to controls (Figure 1A). In females, none of the added BF concentrations altered the food intake when compared to the control group (Figure 1D).

Accordingly, for both sexes the calculated energy content of the ingested food increased in dependence of the dietary fat content (Figure 1B,E). When compared with the corresponding control groups, the calculated energy intake value for males was almost 20% enhanced in the 12% BF group (*p* < 0.001; Figure 1B), whereas for females the respective increase was approximately 150% (*p* < 0.001; Figure 1E). The body weight of male flies was not affected by any of the BF-based HFDs (Figure 1C), while female flies exhibited an even reduced body weight at higher HFD-BF concentrations (e.g., controls = 1.18 mg/female fly versus HFD-12 = 1.09 mg/female fly; Figure 1F), however, without reaching statistical significance. Moreover, irrespective of the high energy intake, neither male (Figure 1G–I) nor female flies (Figure 1J–L) that received the HFDs (HFD-3 to HFD-12) differed significantly in terms of their body weights and body composition. 

### 3.2. Female Fecundity Decreased in a Dose-Dependent Manner in Response to Butterfat Supplementation

We next examined whether the increasing energy intake of female fruit flies that were fed the BF-supplemented HFDs affects the reproduction. As shown in Figure 2A, dietary BF significantly reduced the number of eggs laid per female over a 15-day investigation period in a concentration-dependent manner when compared to control females. The reduced egg laying rate after feeding BF was associated with a significant reduction in the abdominal width of females on day 15 of receiving an HFD-12 as compared to control-fed flies (0.41 ± 0.01 mm for controls versus 0.38 ± 0.01 mm for HFD-12 females; *p* < 0.01) (Figure 2B). Additionally, a significant decrease in ovarian area was observed in comparison to control flies (0.49 ± 0.02 mm^2^ for controls versus 0.43 ± 0.02 mm^2^ for the HFD-12 group; *p* < 0.001) (Figure 2C).

### 3.3. Delayed Egg to Adult Development at High Dietary Butterfat Concentrations

When examining the impact of increasing concentrations of dietary BF on the development of fruit flies, a reduction in the number of developed pupae was observed on an HFD-9 and HFD-12 on day 5 (Figure 2D). Moreover, a significant delay in development time was also visible for the HFD-12 group when the number of hatched flies was counted on day 10 (Figure 2E). However, no differences in terms of percentages of developed pupae and hatched flies compared to the initial egg number, as well as the pupae size between the HFD-groups and controls were observed over the whole observation period (data not shown).

### 3.4. Increasing Concentrations of Dietary Butterfat Did Not Affect the Spontaneous Locomotion Activity of Male and Female Fruit Flies

In male and female fruit flies, a short-term exposure to increasing concentrations of supplemented BF for five days did not significantly affect the spontaneous 24 h locomotion activity when compared to controls (Figure 3A,B). 

### 3.5. Dietary Butterfat Shortened the Lifespan of Both, Male and Female w^1118^ Flies in a Concentration-Dependent Manner

Lifelong ingestion of BF-based HFDs significantly reduced the median, mean and maximum lifespan of male and female flies in a concentration-depended manner when compared to controls (Figure 4 and Table 1). The effect was more pronounced in females than in males. Accordingly, supplementation of 12% (*w*/*v*) BF had the strongest impact reducing the median lifespan of females and males by 23% and 11%, respectively. 

### 3.6. Different High-Fat Diet Qualities Led to Fat Source-Specific Shifts in the Fatty Acid Profile of D. melanogaster

The fatty acid analysis of the supplemented fats and oils indicated that BF with a share of 71% is primary composed of SFAs, such as palmitic, oleic and myristic acids and contains in addition 26% MUFAs and 3% PUFAs (Figure 5A,B). In contrast, SO and LO mainly exhibit PUFAs (SO: 55%, LO: 67%), MUFAs (SO: 34%, LO: 22%) and small proportions of SFAs (11%) (Figure 5A). In particular, SO mainly comprises fatty acids linoleic acid and oleic acid (Figure 5C), whereas LO primarily consists of linolenic acid and to a smaller proportion of oleic and linoleic acid (Figure 5E). OO exhibits predominantly MUFAs (80%) with oleic acid as main component, while amounts of SFAs (13%) and PUFAs (7%) are relatively small (Figure 5A,D). FO in comparison consists of 45% PUFAs and approximately the same proportions of SFAs (31%) and MUFAs (24%). The three major fatty acids of FO are palmitic acid, eicosapentaenoic acid (EPA) and docosahexaenoic acid (DHA) (Figure 5F).

For reference, we first recorded the fatty acid profiles of *D. melanogaster* following feeding on SY control food for 5 days after eclosion. As shown in Figure 6A,B, they were composed primarily of SFAs (males: 48.7 ± 2.6%, females: 45.5 ± 2.5%), MUFAs (males: 41.1 ± 1.6%, females: 46.3 ± 1.6%) and a small proportion of PUFAs (males: 10.2 ± 1.4%, females: 8.1 ± 1.4%). We next examined to what extent a supplemented fat or oil influenced these FA profiles. In most cases, we found that the fat quality of an HFD had a strong impact on the FA pattern of fruit flies (Figure 6). For example, dietary HFD-OO containing predominantly MUFAs led to an increased MUFA content in female flies (HFD-OO: 58.5 ± 2.0%; *p* < 0.001) concurrently reducing the SFA content (HFD-OO: 33.8 ± 1.8%; *p* < 0.01), while males responded with an increased PUFA level (HFD-OO: 14.9 ± 0.4%; *p* < 0.05) (Figure 6A,B). In addition, ingestion of a PUFA-rich HFD-SO and HFD-LO induced a significant increase of the PUFA content in males and females under reduction of MUFAs. Likewise, the SFA content was reduced in female flies (Figure 6A,B). Ingestion of the HFD-LO containing high levels of α-linolenic acid led to a strong increase thereof in both male and female flies (Figure 6C,D). Most notably, this was not true for HFD-FO, which is rich in EPA and DHA. However, its supplementation did not result in an increase of these fatty acids in the fly.

### 3.7. High-Fat Diet Feeding Resulted in an Increased Energy Intake Independent of the Fat Quality

Female fruit flies displayed a significant increase in food intake compared to controls when provided with HFD-SO, HFD-OO, and HFD-LO (Figure 7D). The energy intake per fly was calculated on the ingested food amount and the average energy density of the respective diets. In males, the energy intake was significant higher in all groups that consumed an HFD, compared to the control group (Figure 7B). Likewise, energy intake was significantly increased in females that received an HFD compared to control-fed flies (Figure 7E). Next, we exemplarily examined the impact of an HFD on the consistency of the intestinal excretions. Fruit flies that ingested an HFD-BF, HFD-SO and HFD-FO were found to produce fatty/oily fecal spots (Figure 7M,N).

### 3.8. Body Weight and Body Composition Were Altered by Ingestion of High-Fat Diets Comprising Different Fat Qualities Solely in Female Fruit Flies

However, despite of the increased energy intake (Figure 7B), the body weight and body composition of males that received an HFD-BF, HFD-SO, HFD-OO, HFD-LO and HFD-FO, respectively, did not significantly differ in comparison to males that received the control food (Figure 7C,G–I). In contrast, in females the body weight was significantly reduced by 10% on average when flies were fed an HFD-BF, HFD-LO or HFD-FO (*p* < 0.05; Figure 7F), even under higher energy intake for all HFDs (Figure 7D). Additionally, ingestion of HFD-SO and HFD-LO significantly increased whole body TAG and protein levels in female adults in comparison to an intake of control food (Figure 7J,K).

### 3.9. High-Fat Diets Based on Animal Fats Had a Negative Impact on the Egg Laying Rate in D. melanogaster

The cumulative number of laid eggs per female fruit fly within 15 days post mating was significantly reduced only by ingestion of an HFD-BF and HFD-FO, which represent the animal fat sources in our experimental design (Figure 8A). Although not statistically significant, a slightly decreased number of eggs were counted also on HFD-OO and HFD-SO. In contrast, HFD-LO did not interfere with egg laying.

### 3.10. An Olive Oil Based High-Fat Diet Was Preferred for Oviposition

We next asked whether female fruit flies discriminate between different high-fat diets with respect to egg laying. As shown in Figure 8B, female fruit flies significantly preferred the HFD-OO for oviposition compared to standard food medium with PI values of −0.02 for control versus 0.48 for HFD-OO (*p* < 0.05). Similar trends, although not statistically significant, were obtained for the other plant oils. In contrast, both animal fats, namely BF and FO, were found to be neutral. 

### 3.11. High-Fat Diets Slightly Delayed the Development of D. melanogaster without Affecting the Developmental Rate 

When testing the effect of the different HFD qualities on the egg to adult development of *D. melanogaster*, we found a reduction in number of pupae on day 5 following the ingestion of all HFDs. However, this temporal delay in pupae development was significant solely when eggs were seeded on an HFD-FO (Figure 8C). Moreover, this delay was compensated on the following days of development and the overall development time of flies on day 12 (hatched flies) was not affected by any of the experimental diets (Figure 8D). Neither the percentage of developed pupae, nor the percentage of hatched flies and pupae size was significantly altered when eggs were seeded on an HFD-BF, HFD-SO, HFD-OO, HFD-LO or HFD-FO in comparison to eggs that were seeded on standard food medium (data not shown).

### 3.12. The Fat Quality of High-Fat Diets Affected the Degree of Lifespan Reduction in Fruit Flies

Lifelong ingestion of an HFD significantly decreased the lifespan of male and female *D. melanogaster* irrespective of the added fat source (Figure 9A,B and Table 2). Compared to controls, the median lifespan of males was shortened by 29–44%, and, the mean lifespan by 24–36%. Additionally, the maximum lifespan of male flies was reduced by 9–25%, compared to controls (Figure 9A). The effect of lifelong ingestion of an HFD was even more pronounced in female flies. Here, the median lifespan declined by 63–84%, in comparison to the control group, the mean lifespan by 56–82% and the maximum lifespan by 27–56% (Figure 9B).

Although all fat sources given at 12% (*w/v*) to the diet exhibited a negative effect on the lifespan of fruit flies, the degree of reduction highly depended on the fat quality and the sex of the flies. In males, HFD-SO, HFD-OO and HFD-LO had more severe detrimental effects on lifespan than HFD-BF (*p* < 0.001; *p* < 0.0001; *p* < 0.0001) and HFD-FO (*p* < 0.05; *p* < 0.01; *p* < 0.001). In females, HFD-OO has the strongest detrimental impact on lifespan followed by HFD-SO (*p* < 0.0001), HFD-LO (*p* < 0.01), HFD-FO (*p* < 0.0001) and HFD-BF (*p* < 0.0001), respectively.

### 3.13. High-Fat Diets Affected the Climbing Activity of Fruit Flies in a Fat Quality-Dependent Manner 

Short-term feeding of an HFD for three days had no or only minor effect on the climbing activity of male flies. In comparison to control-treated flies, climbing was significantly reduced solely following the ingestion of the HFD-BF by 15% (*p* < 0.05) and HFD-SO by 14% (*p* < 0.05; Figure 9C). The climbing activity was more drastically diminished in female flies by HFD ingestion, as the intake of the HFD-SO significantly reduced climbing ability by 63%, HFD-OO by 87%, HFD-LO by 70% and HFD-FO by 68% compared to the control (all *p* < 0.001; Figure 9D). Remarkably, feeding the HFD-BF did not affect the climbing rate in females.

### 3.14. The Spontaneous Locomotor Activities of Male and Female Fruit Flies Were Differently Affected by Short-Term Exposure to Plant Oil-Based High-Fat Diets

The spontaneous 24 h activity of 5-day-old male fruit flies was significantly higher compared to females of the same age with 3.38 ± 0.63 count/min/male fly versus 0.54 ± 0.26 count/min/female fly (*p* < 0.001). The locomotion of males was increased when they were fed an HFD-OO and HFD-LO for 3 days (*p* < 0.05; Figure 10A). In HFD-LO male flies the higher spontaneous 24 h activity can be attributed mainly to a higher activity at night-time, whereas in HFD-OO fed males an increased activity at day- and night-time was observed compared to the controls (*p* < 0.05; Figure 10C,E). On the contrary, females displayed a significant impairment of spontaneous 24 h activity following intake of an HFD-OO (*p* < 0.05; Figure 10B). The spontaneous daytime activity in females was reduced following the ingestion of an HFD-SO, HFD-OO and HFD-LO in comparison to the controls (*p* < 0.05; Figure 10D,F).

### 3.15. Fat Quality Had a Large Influence on the Transcriptional Response of D. melanogaster to a High-Fat Diet 

Whole-body transcriptome analysis of female *D. melanogaster w^1118^* was performed following a three-day ingestion of an SY control, HFD-BF, HFD-SO and HFD-FO, respectively. A total of 17,697 genes were investigated of which 670, 342, and 500 genes were found to exhibit a significantly changed transcript level after ingestion of an HFD-BF, HFD-SO and HFD-FO, respectively, when compared to SY controls (Appendix A). In the HFD-BF group the expression of a larger number of genes was upregulated (57%) than downregulated (43%) (Figure 11A). Intake of an HFD-SO resulted in an upregulated expression of 52% and a downregulation of 48% of identified genes (Figure 11B). However, ingestion of an HFD-FO, primarily downregulated the expression of genes (65%) rather than elevated gene expression (35%) (Figure 11C). 

When using DAVID bioinformatics software, we retrieved functional annotation clusters of HFD-regulated genes (Table 3; Appendix A). However, several significantly regulated genes have not yet been functionally annotated in the DAVID database. Hence, we performed a literature search on flybase and PubMed to obtain recent information on these genes and, if appropriate, to extend the functional clusters. If at least two genes were identified that were not part of any DAVID cluster but based on their biological function could be clustered, they form orphan categories. Among the 39 genes that were significantly upregulated in female *D. melanogaster* by the HFD-BF, 28 genes are related to the functional cluster “innate immunity/stress response”. The same functional cluster also came up when analyzing the transcripts that were upregulated by an HFD-FO (5 genes). In addition, three genes matching the term “monooxygenase activity” were significantly enriched in the HFD-FO group. Via eye inspection, genes that are involved in “fatty acid metabolism/fatty acid elongation” were identified in all HFD groups. Three “alkaline phosphatase” genes were found to be present among the upregulated transcripts of HFD-FO flies.

For the significantly downregulated genes, functional clusters were retrieved only among the transcripts of HFD-FO fed flies, namely, “proteolysis” (11 genes), “lipase activity” (4 genes), and “odorant binding” (3 genes). 

### 3.16. The Small Overlap of Genes Differentially Expressed in Response to All of the Three High-Fat Diet Qualities Were Associated with the Functional Terms Immune/Stress Response and Fatty Acid Metabolism

Remarkably, a high proportion of the significant transcriptional changes were found to be fat quality-specific. For HFD-BF the percentage value was 72.5%, for HFD-FO 60.6%, and for HFD-SO 47.0% (Figure 11D). Accordingly, the overlaps of significantly regulated genes between the three fat qualities are relatively small (Figure 11D). The transcript levels of the 13 genes of the HFD-BF/HFD-SO overlap are equally regulated (Appendix A). However, they did not form a functional annotation cluster, although 3 of the 13 genes were predicted to be involved in proteolysis. Genes that are common between females fed an HFD-BF and HFD-FO, respectively, and maintained similar directionality of the fold-change cluster in the terms “innate immune response” and “peptidases”. In total, 5 of the 42 genes exhibited an opposite regulation (Appendix A). The overlap of significantly changed genes between the HFD-SO and HFD-FO feeding groups was the largest. Of the 101 genes, only 1 gene was oppositely regulated (Appendix A). Five functional clusters were retrieved, namely “proteolysis”, “lipase activity”, “regulation of Notch signaling pathway”, “alkaline phosphatase” and “regulation of transcription from RNA polymerase II promoter”.

Only 30 genes were identified whose expression was modified by the ingestion of all three HFDs (HFD-BF, HFD-SO, HFD-FO) as compared to the control group. Among those 30 genes, the expression of 17 genes was generally upregulated and that of 12 genes downregulated, while one gene, CG14499 was inconsistently regulated (Figure 11E; Appendix A). Functional cluster analysis revealed an enrichment of upregulated genes involved in “innate immunity/stress response” and “fatty acid metabolism (fatty acid elongase activity/sphingolipid biosynthetic process/biosynthesis of unsaturated fatty acids”. In particular, ingestion of either HFD-BF, HFD-SO, or HFD-FO induced the expression of turandot A, M, X, C (*TotA, TotM, TotX, TotC*), peptidoglycan recognition protein SB1 (*PGRP-SB1*), and galactose-specific C-type lectin (*lectin-galC1*), all of them participating in the innate immune and stress response (Figure 11E). In addition, a literature search revealed that the upregulated gene CG18585 encodes a putative zinc-dependent carboxypeptidase similar to mammalian carboxypeptidase A1, which in humans is involved in the eicosanoid biosynthesis pathway [46]. The fatty acid metabolism cluster retrieved by DAVID bioinformatics software is represented by the upregulated genes CG18609, CG30008, and CG9458, encoding putative orthologues to the human fatty acid elongases ELOVL1 and ELOVL7, as well as by the stearoyl-CoA 9-desaturase Fad2. Moreover, our literature and flybase search revealed the ω-hydroxylase Cyp4g1 [47], CG17562 encoding a predicted alcohol-forming fatty-acyl-CoA reductase and CG7910 that is annotated as an acylglycerol lipase/fatty acid amide hydrolase. Taken these literature data into account, only 2 of the 17 upregulated genes do not fit to the two clusters. No functional cluster was found for the 12 downregulated genes. 

### 3.17. Feeding an HFD Containing 12% Butterfat Had a Negative Impact on Gut Integrity 

Next, we exemplarily investigated for HFD-BF feeding, whether the elevated expression of genes involved in immune function may be an indicator of intestinal stress owing to an impaired gut integrity. As shown in Figure 11G, populations of females flies were found to exhibit a significantly higher proportion of individuals with the smurf phenotype when they were exposed to HFD-BF as compared to controls. 

## 4. Discussion

In the current study, we have comparatively investigated the impact of short- and long-term feeding of different dietary fat quantities and qualities on the model organism *D. melanogaster*. BF, which is characterized by a high content of SFAs (71% of the total fat), such as palmitic acid, oleic acid and myristic acid, was chosen to evaluate the effect of an increasing dietary fat content on the health status of *D. melanogaster*. Although the calculated energy intake drastically increased in a dose-dependent manner, supplementing BF did not alter the body composition and body weight at any concentration tested. In line with this, we observed fatty/oily fecal spots when fruit flies were exposed to HFD, which indicates that most of the ingested fat is excreted unprocessed. The majority of previous HFD studies in *D. melanogaster* used coconut oil as fat source, which contains also mainly SFA (82.5% of the total fat), however in a different composition, namely lauric acid, myristic acid and palmitic acid. In contrast to our results, these feeding studies usually reported increased TAG levels in male and female *D. melanogaster* in response to HFD feeding [19,21,26,48,49], which were shown to rise in a dose-dependent manner and were already manifested at low dietary fat concentrations of 3–5% [21,23]. However, when monitoring the impact of a coconut oil-based HFD on several metabolic parameters over time, Liao et al. [20] found a dynamic progression of the TAG content. It did not differ between HFD and control animals at day 5, displayed an increase for HFD fed flies at time point day 7, and, remarkably, was even lower in flies on HFD than in flies on control diet after three weeks. Accordingly, we suggest that the dose-dependent negative effects on lifespan, fecundity and larval development by dietary BF are not attributable to an obese phenotype. These life history traits were found to be similarly impaired by coconut oil- and lard-based HFD [19,20,23,25,27,28]. Remarkably, the decreased lifespan of *D. melanogaster* in response to a 12% BF HFD was associated with a higher proportion of animals showing a disturbed gut integrity. This finding is in line with studies by Rera et al. [35] who demonstrated that intestinal barrier dysfunction is an important proxy for the probability of impending death and, hence, lifespan. 

When we comparatively investigated the impact of other dietary fat qualities that considerably differ in their FA composition ranging from high PUFA (HFD-SO, HFD-LO, HFD-FO) over high MUFA (HFD-OO) to high SFA (HFD-BF) proportions on the food uptake rate and the TAG level of fruit flies, we obtained diverse results. None of the supplemented fats/oils influenced the TAG content and the body weight of male fruit flies, although the food intake and concurrently the calculated energy intake were significantly increased by all HFDs. In contrast to that, female fruit flies that were fed the plant derived oils SO and LO exhibited significantly enhanced TAG levels, which were accompanied by an increased food intake. The animal-derived BF and FO had only a minor, non-significant effect on food uptake in females and did not alter their TAG content despite a markedly enhanced energy content in the ingested food. Food consumption of fruit flies was also found to be increased by a coconut oil supplemented HFD [20,48], which is in line with the situation in rodents and humans, where HFDs promote hyperphagia [50]. Similar to previously outlined findings with a 20% coconut oil-based HFD [20], ingestion of an HFD decreased the body weight in females, which is most probably related to the reduction in egg production discussed below. 

In good agreement with our BF data and the already abovementioned coconut oil- and lard-based HFD studies [19,20,23,25,27,28], we found that the consumption of an HFD had a negative impact on the lifespan of *D. melanogaster*. Although this was true for all of the tested fats/oils, a clear impact of the fat quality on the degree of reduction was observed. Male and female fruit flies that consumed the plant oils OO, SO and LO exhibited a shorter lifespan when compared to animal-derived BF and FO. Epidemiological studies and human clinical trials indicate that a diet rich in MUFAs, such as the Mediterranean diet, with a high OO consumption rate decreases the risk of metabolic syndrome and cardiovascular disease and, hence, promotes health span [7]. In line with this, when tested under non-HFD conditions, individual MUFAs (oleic acid, palmitoleic acid or cis-vaccenic acid) supplemented at a concentration of 0.8 mM (<0.05%; *w*/*v*) were found to prolong the lifespan of the model organism *Caenorhabditis elegans* [51]. Similarly, lifespan extensions were obtained when 10 µM ω-6 PUFAs (arachidonic ac-id; di-homo-γ-linoleic acid) (<0.0005%; *w*/*v*) were individually added to the nematode medium [52]. Therefore, it would be interesting to perform similar non-HFD supplementation studies in *D. melanogaster* using a chemically defined medium [18].

The aging process of *Drosophila* is accompanied with a decrease in physical activity [53]. Consistently, the shortening impact of an HFD on longevity was reported to be accompanied with an impaired negative geotaxis climbing activity [19,20,21,22,23]. However, we again obtained a differentiated picture, when we tested the effect of a short-term exposure to HFD consisting of different fat qualities on the climbing ability of fruit flies. In males, short-term treatment with BF and SO slightly decreased the induced locomotor activity, while the other fats/oils did not have an effect. In contrast, female flies were found to be more sensitive towards an HFD except for BF (the fat quality with the lowest impact on aging), resulting in drastically reduced climbing activities. Similarly, we revealed distinct effects of the different HFD qualities on the spontaneous locomotor activity of fruit flies. In males, the plant oils OO and LO led to increased locomotion, whereas in females, all HFD qualities, especially the plant derived oils, had a negative impact on the spontaneous physical activity. Similarly, supplementation of 30% coconut oil for one week has been reported to lead to a less active phenotype in virgin fruit fly females [20]. The sex-specific discrepancy in the spontaneous locomotor activity response to an HFD awaits further investigation. 

A lowered egg production in response to an HFD was revealed in our studies for all fat qualities except LO. This confirms a previous feeding study which reported a negative impact of a coconut oil-based HFD on the fecundity of female *D. melanogaster* [20]. Maternal obesity has been frequently shown to interfere with egg production and female fertility in *D. melanogaster* [54] and in other animals, including mammals [55]. However, since our HFD feeding protocol did not generally lead to an enhanced TAG level in female fruit flies, we suggest that the egg production decreased independently of an obese phenotype. Remarkably, in mice an elevated dietary fat intake was found to be associated with impaired estrous cycle, depletion of the ovarian reserve, and transcriptional changes of genes involved in the ovulation process, regardless whether an obese phenotype was or was not induced [55]. 

Moreover, we found that the larval development of fruit flies was slightly delayed when eggs were placed on HFD of different fat quality with FO exhibiting the strongest impact. The observed developmental delay may be caused by the proportional reduction of the carbohydrate and protein content by elevating the fat content of the diet following the concept of the “geometry of the diet” [56]. Accordingly, an imbalance of macronutrient intake, in particular of the sugar to protein ratio, is known to markedly affect the developmental timing of *D. melanogaster* [57]. 

Considering the mainly negative effects of SO, OO and LO, it is quite remarkable that female fruit flies were found to prefer diets supplemented with these plant oils as egg laying sites. In contrast, animal derived fats/oils had no influence on oviposition. Fruit flies are able to taste and discriminate between dietary FA [58]. Hence, differences in the content and quality of free FA among the tested fats and oils may be responsible for choice of the egg laying site. However, we cannot rule out that other ingredients, such as plant secondary metabolites, that are present in very low concentration in plant oils may act as attractants.

Of note, diverse responses to HFDs of different fat qualities were also revealed when we analyzed the differential gene expression in female fruit flies. In HFD-BF-fed flies, the majority of the differentially expressed genes were up-regulated, whereas the HFD-SO group exhibited an almost balanced number of up- and down-regulated genes. In contrast, HFD-FO feeding favors the downregulation of genes. Consistently, only in the latter group an enrichment analysis revealed functional clusters among the downregulated genes. The suppression of genes involved in proteolysis (among others, four trypsin and four serine protease genes), and lipolysis (the triacylglycerol lipases CG17192, CG6295 and CG6283) may indicate an adaptation to the surplus of energy intake by the dietary FO. In line with this, we also identified eight downregulated genes involved in glycosidase/glucose transport (two α-amylases, the α-glucosidases *sug* and *Mal-2A*, the trehalase CG6262, the putative hexose transporters CG32054, CG17929 and CG6901; however, they no longer reached a significant level of *p* < 0.1 after FDR correction). Remarkably, similar to our HFD-BF results, a 20% coconut oil-based HFD has been repeatedly shown to lead to more upregulated than downregulated genes [48,59,60]. Both, BF and coconut oil are characterized by a high proportion of SFA. In this regard, it is of note that the number of genes exclusively regulated by the two fat sources rich in PUFA (FO and SO share 101 significantly regulated transcripts) is considerably higher than the exclusive overlaps of the two animal derived fats FO and BF with 42 common genes and that of SO and BF with only 13 genes.

For all three HFD groups, we revealed an upregulation of transcripts associated with the immune/stress response of *D. melanogaster*. It is well established that the aging process and age-related diseases are accompanied by an activation of inflammatory pathways and immunity-related genes in *Drosophila* and also in humans [35]. Hence, our transcript data are consistent with the observed general shortening of fruit fly lifespan by an HFD no matter what fat quality was used. However, most of these immune/stress response genes (fold change > 1.5, FDR < 0.1) were found to be specifically induced, which is shown by the significantly different number of affected genes (28 genes for HFD-BF, 2 genes for HFD-SO, 6 genes for HFD-FO). In this regard, it is of note that in HFD-FO flies, we identified four differentially expressed alkaline phosphatase genes (of which CG3292 *alp7* does not reach significance level after FDR correction). The function of Alp in *D. melanogaster* has not yet been studied. However, homologues of vertebrates have been demonstrated to play a vital role in maintaining gut homeostasis and shaping the gut microbiota. For instance, vertebrate intestinal alkaline phosphatases whose expression is induced in the presence of bacteria have an anti-inflammatory function by inactivating LPS [61]. An enrichment of genes functioning in immunity, response to infection and stress was also reported for the upregulated transcripts of female fruit flies when 20% coconut oil HFDs were used [48,49,59,60]. Stobdan et al. [48] compared the gene expression in the heads and bodies of male and female *D. melanogaster* that were altered by an HFD and retrieved only four genes that were significantly upregulated in all groups, namely *lsp2*, *gnmt*, *cyp4e3* and CG5953. *gmnt* encoding the glycine C-acetyltransferase was the most upregulated transcript in fruit flies that were fed an HFD-FO and was found to be also 4-fold (however, after FDR correction no longer significantly) upregulated in HFD-SO flies, while it did not appear among the significantly regulated transcripts in the HFD-BF group. Expression of *D. melanogaster gnmt* was found to be elevated in response to inflammatory Toll activation, and genetic studies indicate that Gnmt function is crucial for maintaining a proper TAG content in the fruit fly [62]. *lsp2* encoding the Larval serum protein 2 exhibited a significant +8.5 fold increased expression in the HFD-FO flies but was not found to be differentially expressed in the two other HFD groups. The role of Lsp2 whose expression is regulated by the steroid hormone 20-hydroxyecdysone (20E) in the adult flies has not yet been unveiled. The transcript level of *cyp4e3* and CG5953 were not changed by any of our HFD. In a previous 7 day feeding study using *w^1118^* females, Heinrichsen et al. [59] reported that an HFD supplemented with 20% coconut oil suppressed the expression of an argininosuccinate lyase (ASL) homologue and genetic downregulation by RNAi was sufficient to induce an obese phenotype, whereas overexpression rescued a cold tolerance phenotype and the TAG level of flies on HFD. Moreover, Hemphill et al. [60] found among the transcripts that were downregulated by the 20% coconut HFD an enrichment of genes related to hemocyanin. Both, *asl* and hemocyanin genes were not differentially regulated by any of our HFD treatments. It is important to note here that differences in transcriptome data may be due to the fact that we used a lower lipid concentration of 12% compared with the previous studies using HFD containing 20% coconut oil.

One possibility of how the different fat sources provoked the diverse responses in *D. melanogaster* is their impact on the fruit fly’s fatty acid profile. We found that the intake of an HFD led to fat source-specific shifts in the fatty acid pattern of *D. melanogaster*, implying an efficient absorption of the dietary FAs. This may be relevant in the context of the production of edible insects used as food and feed [63] because it allows one to modulate the FA pattern of the insects via the choice of an appropriate dietary fat source. A remarkable exception was the long chain unsaturated FAs DHA and EPA (C20 and C22), which are present in high concentrations in FO. However, an HFD supplemented with FO did not increase the usually very low levels of these FA in *D. melanogaster*. Therefore, we suggest that they are either not taken up or rapidly metabolized by the fruit fly. A rapid conversion of dietary C22 PUFAs given as free FAs to shorter FAs by the fruit fly has been previously reported [64]. *D. melanogaster* does not possess enzymes to synthesize linoleic acid and α-linolenic acid *de novo*. In addition, it lacks, contrary to mammals, δ-5 and δ-6 desaturases and thereby cannot synthesize long-chain C20 and C22 PUFAs (namely arachidonic acid, EPA and DHA) from dietary linoleic acid and linolenic acid [64]. Considering the assumption by Carvalho et al. [65] that *D. melanogaster* is capable to synthesize all FAs de novo that are required for life, C20 and C22 PUFAs may not be required for lipid membrane formation or for the synthesis of, e.g., eicosanoids in the fly. Our RNAseq data revealed a small overlap of transcripts upregulated by all three fat qualities tested, which exhibits an enrichment of genes involved in FA metabolism, including putative orthologues of human fatty acid elongases, the stearoyl-CoA 9-desaturase Fad2 involved in the biosynthesis of unsaturated fatty acids and a predicted alcohol-forming fatty-acyl-CoA reductase, thereby putatively participating in the long-chain fatty-acyl-CoA metabolic process. The oenocyte-specific ω-hydroxylase cytochrome P450-4g1 (*cyp4g1*) has been demonstrated to be a key regulator of the FA composition of TAG in *D. melanogaster* [47]. Moreover, the predicted acylglycerol lipase/fatty acid amide hydrolase CG7910 may support an FA catabolic process. Together, this may indicate an increased processing and remodeling of the ingested FAs. 

In humans, it is well established that the quality of the dietary FA intake modulates the FA pattern of cellular TAG and phospholipids with influences on metabolic processes and health parameters, since FA are not only energy sources but also serve as signaling molecules regulating distinct physiological functions and cellular processes [66,67] A good example of this is the beneficial effect of EPA and DHA consumption on thermogenesis, inflammation and cardio-metabolic diseases [66]. Moreover, the fat quality (the length of the carbon chain and the saturation state) was found to be an important parameter that determines how the intestinal microbiota and the immune system are affected by dietary fats [67]. Consistently, epidemiological studies unveiled a correlation between fat qualities and health risks [5]. In *Drosophila*, dietary lipids have been reported to elicit structural and functional changes in the intestinal tract. In particular, they positively influenced enteroendocrine cell differentiation via Notch signaling [68]. On the other hand, a recent study demonstrated that an HFD containing 20% (*w*/*v*) palm fat (50% SFA, 40% MUFA) alters the microbiota community and increased the bacterial load of the fruit fly intestine, which leads to a transient activation of the intestinal stem cells [69]. Along with c-Jun N-terminal kinase-dependent activation of cytokine expression, intestinal cell number and cellular composition are modified. It would be interesting to examine whether these effects are influences by the fat quality of the applied HFD. The data by von Frieling et al. [69] are consistent with the impact of dietary factors, such as an HFD, on the complex interaction between host and microbiota, which affects intestinal morphology, immune function and gut health in *D. melanogaster* and likewise in mammals [67,70]. Hence, one can speculate that fat quality may influence the alterations of the composition and activity of the intestinal microbiota elicited by an HFD, which in turn differentially affects the gene expression patterns in *D. melanogaster*. 

Given that an HFD increased the bacterial density in the intestine [69] and that an enhanced bacterial load can elicit an immune response [70] and may be responsible for the intestinal barrier dysfunction observed in the present study, it fits that our transcriptional analyses of HFD fruit flies unveiled a bunch of upregulated genes that are involved in the immune and stress response. This finding is also in good accordance with previous HFD studies in *D. melanogaster* [49,59,60]. In mammals, feeding an HFD has been broadly demonstrated to cause dysbiosis of the intestinal microbiota with concurrent activation of stress and immune signaling pathways and changes of gut epithelial cells, which may lead to the development of a leaky gut [71]. Loss of intestinal integrity in association with changes in metabolic and immune signaling can be considered as a predictor of death [35]. The increased expression levels of immune and stress response genes that we observed in HFD fed fruit flies may improve the stress tolerance and mitigate adverse activities of the intestinal microbiota. A good example for this is the *pgrp-SB1* transcript that belongs to the overlap of 30 genes significantly induced by all three fat qualities, although to a different extent (+9.2-fold in HFD-BF, +5.4-fold in HFD-FO and +1.9-fold in HFD-SO treated fruit flies). PGRP-SB1 has been shown to exhibit amidase activity against DAP-type peptidoglucans and, hence, can potentially counteract immune activation [72]. Consistently, the negative impact of an HFD on the lifespan was lower in fruit flies following HFD-BF or HFD-FO ingestion, where a considerably higher number of immune and stress response genes were upregulated, when compared with flies fed an HFD-SO. 

## 5. Conclusions

In the present study, we found that in *D. melanogaster* the fat quality of an HFD (i) is a crucial modifier of the FA profile, (ii) diversely affects the transcriptional response to an HFD and (iii) differentially modulates several life history traits. Moreover, we exemplarily demonstrated for BF-based diets that the effect of an HFD on several of these traits including longevity, egg production and larval development follows a dose-dependency. Irrespective of the fat quality, our transcriptional data indicate that the ingestion of an HFD induces genes involved in the immune/stress response of *D. melanogaster*. Since we exemplarily demonstrate that gut integrity is significantly disturbed by feeding *D. melanogaster* an HFD-BF, this suggests that an HFD may be generally associated with a dysbiosis of the intestinal microbiota, structural alterations and barrier dysfunction of the gut. We cannot exclude the possibility that factors, such as the consistency of the fat source or minor ingredients, may also contribute, at least in part, to the observed effects. However, similar to the situation in mammals, we assume that the FA profile is most probably the crucial factor that determines effect sizes of fats and oils supplemented to fruit fly diets. This should be taken into account when HFD studies with *D. melanogaster* are carried out. 

## Figures and Tables

**Figure 1 cells-11-04043-f001:**
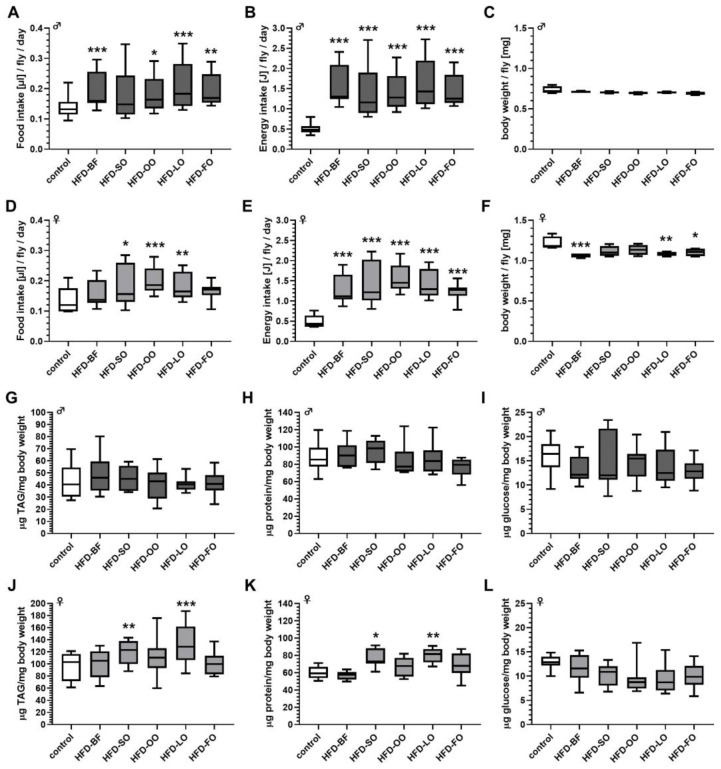
High-fat diets (HFD) comprising 3–12% (*w*/*v*) butterfat (BF) increased the calculated energy intake without affecting body weight and body composition of male and female *D. melanogaster*. *D. melanogaster w^1118^* were fed ad libitum for 7 days on a standard sugar yeast control medium and on HFDs comprising increasing BF contents, respectively, before their food consumption and body composition were examined. (**A**) In males, the food intake was not affected up to 9% BF, however, significantly increased on an HFD-12, (**D**) whereas female flies did not change their food consumption at any applied BF concentration (*n* = 3 replicates; *N* = 225 flies per treatment in total). The corresponding energy content values for the ingested food, which were calculated for (**B**) male and (**E**) females indicate that, in both sexes, the energy content was dose-dependently and significantly increased by BF supplementation. The body weight and body composition (triacylglycerol (TAG), protein, glucose content) of male (**C**,**G**–**I**) and (**F**,**J**–**L**) female flies did not differ between the treatment groups (body weight: *n* = 3 replicates, *N* = 120 animals per treatment; body composition: *n* = 3 replicates; *N* = 18 flies per treatment). Data are shown as boxplots with whiskers indicating minimal and maximal values. Statistical significance was assumed at * *p* < 0.05, ** *p* < 0.01, *** *p* < 0.001 (ANOVA). (HFD-3: high-fat diet with 3% (*w*/*v*) BF, HFD-6: high-fat diet with 6% (*w*/*v*) BF, HFD-9: high-fat diet with 9% (*w*/*v*) BF, HFD-12: high-fat diet with 12% (*w*/*v*) BF).

**Figure 2 cells-11-04043-f002:**
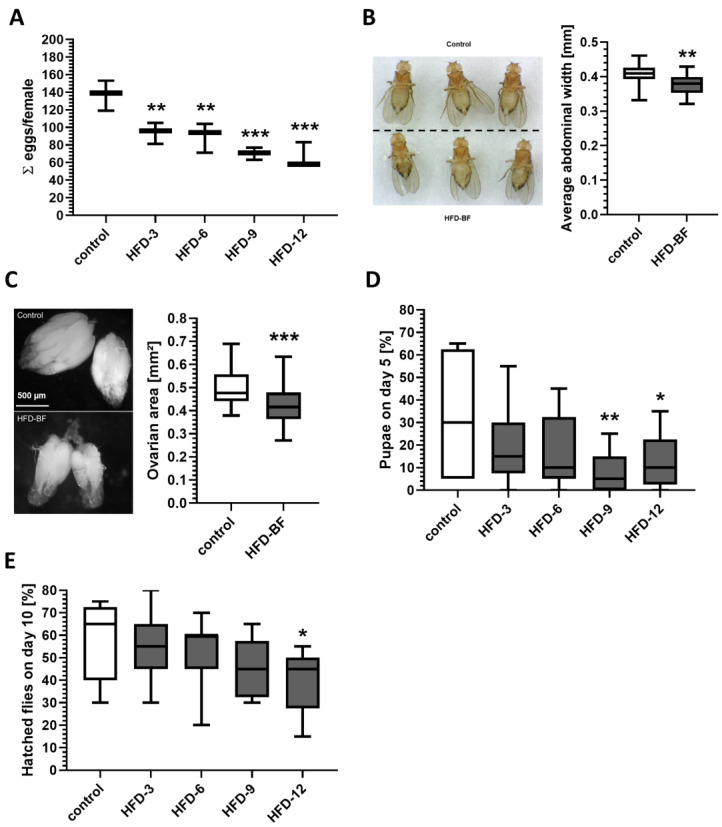
Butter fat (BF) supplementation reduced female fecundity and prolonged the egg to adult development time. (**A**) To examine the impact of increasing concentrations of BF on female fecundity, female *D. melanogaster w^1118^* were placed on standard control medium or HFD containing 3%, 6%, 9% and 12% (*w*/*v*) BF. Flies were maintained under standard conditions throughout experimental treatment and the number of laid eggs per vial was counted over a period of 15 days (*n* = 3 independent experiments with *N* = 18 females per treatment in total). The cumulative number of laid eggs per female was significantly and dose-dependently reduced following ingestion of an HFD. (**B**) Compared to controls, the reduced egg laying rate of HFD-BF females was associated with a decreased abdominal width and (**C**) ovary size when analyzed at day 15. The abdominal width (*n* = 2 independent experiments with *N* = 15 females per treatment) and area of ovaries (*n* = 2, *N* = 40 ovaries per treatment) were quantified using ImageJ. (**D**) To investigate the influence of increasing concentrations of BF on fruit fly development, 20 eggs derived from *w^1118^* females, which were pre-fed with standard food medium were placed on control medium or HFD. The number of pupae and hatched flies per vial were counted for 12 days. A significant time shift in pupae development towards a reduced number of developed pupae out of 20 eggs per vial on an HFD-9 and HFD-12 was observed on day 5. (**E**) A delay of the development time was also visible detected with regard to a reduced number of hatched flies on an HFD-12 on day 10. (For (**D**,**E**): *n* = 3 replicates, *N* = 180 eggs per treatment in total. Data in (**A**–**E**) are shown as boxplots with whiskers indicating minimal and maximal values. Statistical significance was assumed at * *p* < 0.05, ** *p* < 0.01, *** *p* < 0.001: Significant differences between groups, evaluated by ANOVA).

**Figure 3 cells-11-04043-f003:**
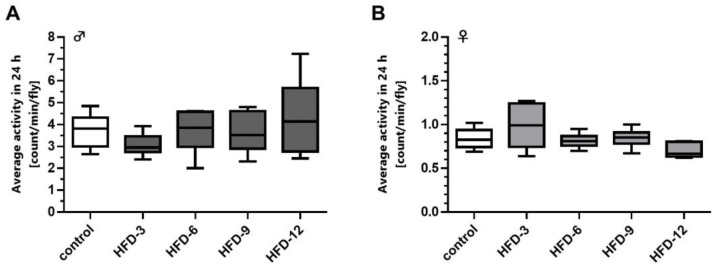
Short-term exposure to high-fat diets (HFD) comprising 3% to 12% (*w*/*v*) butterfat (BF) did not affect the spontaneous locomotion activity of male and female *D. melanogaster. D. melanogaster w^1118^* were fed ad libitum with a sugar yeast control diet or HFD comprising 3%, 6%, 9%, 12% (*w*/*v*) BF for 5 days under standard condition, before the spontaneous activity was recorded with the Drosophila Activity Monitoring system at intervals of 10 min after an adaptation period of 24 h. (**A**) At none of the concentrations applied did BF significantly alter the spontaneous activity in males and (**B**) females (*n* = 3 replicates; *N* = 120 flies per treatment in total. Data are shown as boxplots with whiskers indicating minimal and maximal values. Statistical significance was evaluated by ANOVA. HFD-3: high-fat diet with 3% (*w*/*v*) BF, HFD-6: high-fat diet with 6% (*w*/*v*) BF, HFD-9: high-fat diet with 9% (*w*/*v*) BF, HFD-12: high-fat diet with 12% (*w*/*v*) BF.

**Figure 4 cells-11-04043-f004:**
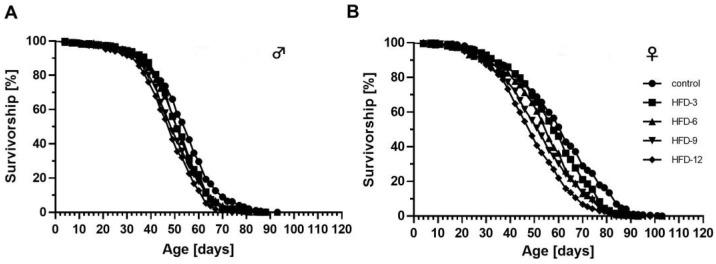
Lifelong ingestion of a high-fat diet (HFD) comprising 3–12% (*w*/*v*) butterfat (BF) reduced the lifespan of male and female *D. melanogaster.* (**A**) HFDs containing increasing concentrations of BF (3%, 6%, 9%, 12% (*w*/*v*)) significantly reduced the lifespan of male and (**B**) female *w^1118^* flies compared to controls (Logrank *p* < 0.001; *n* = 3 with *N* = 450 animals per treatment; a merge of all three lifespans is shown). The corresponding median, mean and maximum lifespan data are listed in Table 1. HFD-3: HFD with 3% (*w*/*v*) BF, HFD-6: HFD with 6% (*w*/*v*) BF, HFD-9: HFD with 9% (*w*/*v*) BF, HFD-12: HFD with 12% (*w*/*v*) BF.

**Figure 5 cells-11-04043-f005:**
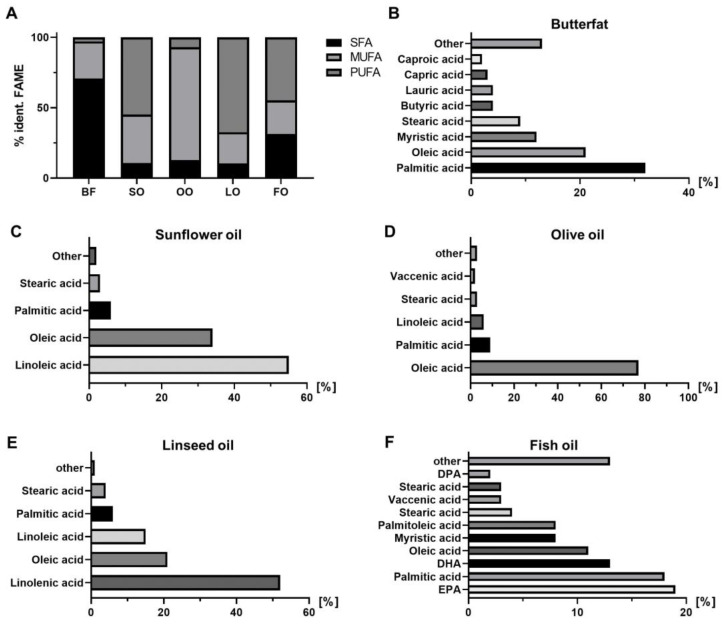
Fatty acid composition of the fats and oils used as supplements in the *D. melanogaster* feeding studies. (**A**–**F**) The fatty acid profiles of fats and oils were determined by capillary gas chromatography of fatty acid methyl esters (FAME) and data are calculated as % identified FAME. Butterfat (BF) predominantly consists of saturated fatty acids (SFAs), sunflower oil (SO), linseed oil (LO) and fish oil (FO) of polyunsaturated fatty acids (PUFAs) and olive oil (OO) of monounsaturated fatty acids (MUFAs).

**Figure 6 cells-11-04043-f006:**
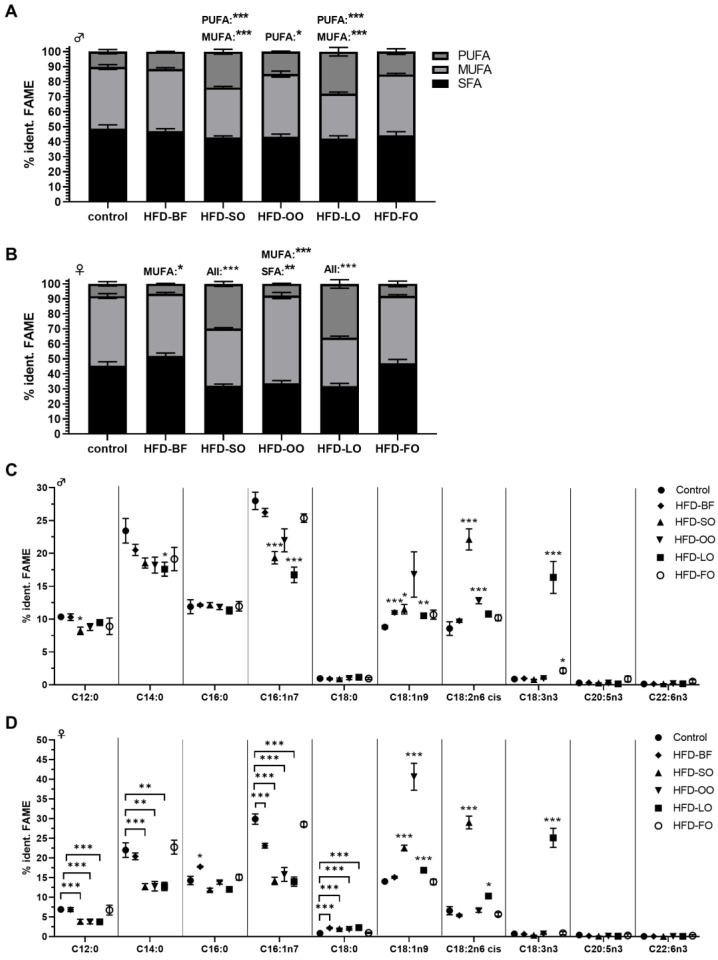
Alteration of the fatty acid (FA) profile of *D. melanogaster* after feeding on high-fat diets (HFD) with different fat qualities. Two-day-old *D. melanogaster w^1118^* were fed ad libitum for 3 days on a standard sugar yeast medium and the different indicated HFDs comprising 12% (*w*/*v*) BF, SO, OO, LO or FO, respectively. (**A**–**D**) The FA profiles were determined by capillary gas chromatography with flame ionization detector. The data are calculated as % identified (ident.) FA methyl esters (FAME) and displayed for (**A**,**C**) males and (**B**,**D**) females (*n* = 3; *N* = 2175 male or *N* = 1350 female flies per treatment in total). Data are shown as means ± SEM. Statistical significance was assumed at * *p* < 0.05, ** *p* < 0.01, *** *p* < 0.001 (ANOVA). HFD-BF: high-fat diet with butterfat, HFD-SO: high-fat diet with sunflower oil, HFD-OO: high-fat diet with olive oil, HFD-LO: high-fat diet with linseed oil, HFD-FO: high-fat diet with fish oil.

**Figure 7 cells-11-04043-f007:**
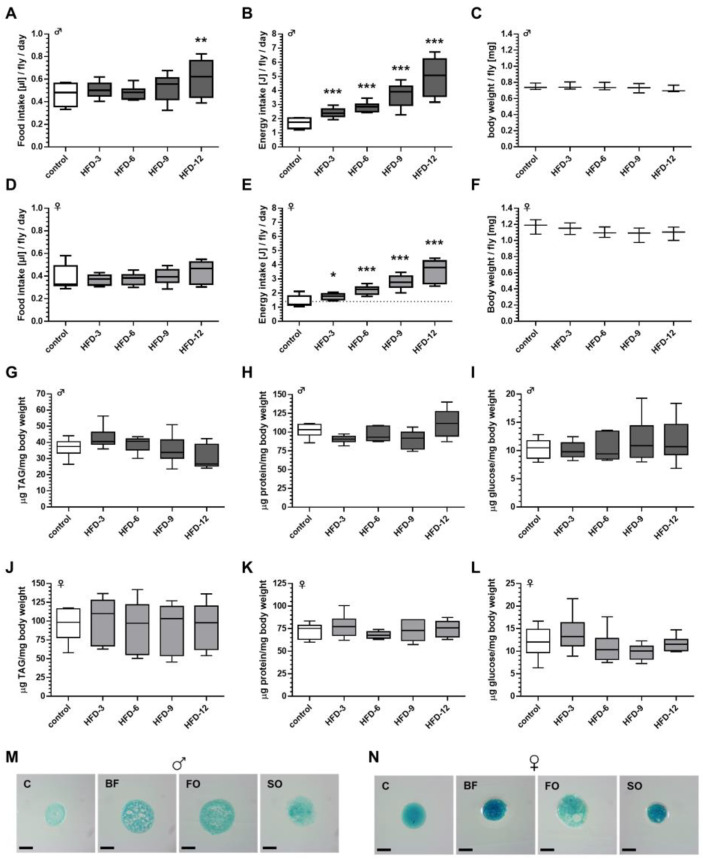
Short-term consumption of a high-fat diet (HFD) with different fat qualities significantly increased the energy intake of both male and female *D. melanogaster* while body composition was solely altered in female flies. *D. melanogaster w^1118^* flies were fed ad libitum with the indicated experimental diets for 5–7 days. Food was renewed 3 times a week and the flies were kept under standard conditions throughout experimental treatment. Food intake was examined by EX-Q. (**A**) In males, the food intake was increased by ingestion of an HFD-BF, HFD-OO, HFD-LO or HFD-FO, (**D**) while in females, it was elevated in response to an HFD-SO, HFD-OO or HFD-LO. The corresponding calculated total energy intake was significantly enhanced in all males (**B**) and females (**E**) that received an HFD (*n* = 3 replicates with *N* = 225 animals per treatment). (**C**) The body weight of males was not altered by any diet, (**F**) while that of females was significantly reduced following intake of HFD-BF, HFD-LO or HFD-FO compared to controls (*n* = 3 replicates with *N* = 120 flies per treatment). (**G**–**I**) Body composition was not affected in male flies by any diet, (**J**–**L**) but was significantly altered in terms of TAG and protein content in females following a 5-day feeding period with HFD-SO and HFD-LO in comparison to control-fed flies (*n* = 3 replicates; *N* = 18 animals in total). Data are shown as boxplots with whiskers indicating minimal and maximal values. Statistical significance was assumed at * *p* < 0.05, ** *p* < 0.01, *** *p* < 0.001: Significant differences between groups, evaluated by ANOVA. HFD-BF: high-fat diet with butterfat, HFD-SO: high-fat diet with sunflower oil, HFD-OO: high-fat diet with olive oil, HFD-LO: high-fat diet with linseed oil, HFD-FO: high-fat diet with fish oil. (**M**) Fecal spots of male and (**N**) female flies were exemplarily captured by microscopy on day 7 after feeding blue-dyed experimental diets. The scale bars indicate 0.2 mm.

**Figure 8 cells-11-04043-f008:**
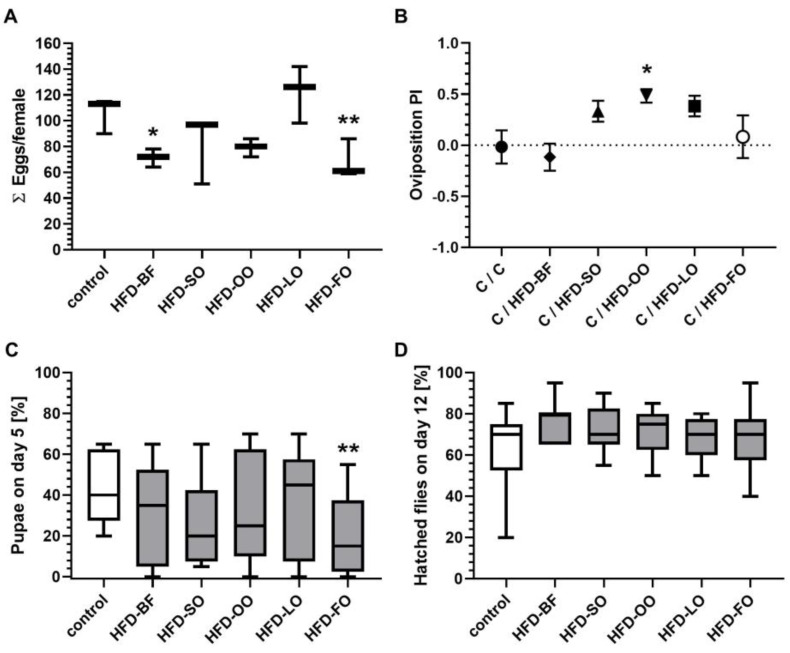
Female fecundity and oviposition as well as egg to adult development were affected by HFD in a fat quality dependent manner. (**A**) To assess female fecundity, two *D. melanogaster w^1118^* females at a time were maintained in vials containing sugar yeast medium (control) or an HFD and the number of laid eggs per vial was counted over a period of 15 days. The cumulative number of laid eggs per female was significantly reduced following ingestion of an HFD-BF and HFD-FO and tended to be lowered by an HFD-SO and HFD-OO (*n* = 3, *N* = 18 females). (**B**) To evaluate oviposition preferences, female fruit flies were pre-fed for 7 days on sugar yeast medium, before they were allowed to choose between two petri dishes containing control food and HFD, respectively. The number of laid eggs per petri dish within 24 h was counted. Females significantly preferred an HFD-OO rather than control food (*n* = 3 replicates with *N* = 150 females per condition). Statistical significance was assumed at * *p* < 0.05 (ANOVA). (**C**) To investigate the influence of HFDs on the egg to adult development, 20 *D. melanogaster* eggs were seeded on control food and HFD enriched with different fat sources, respectively. The number of pupae and hatched flies per vial were monitored over a period of 12 days. The number of pupae on day 5 are depicted. (**D**) The observed slight delay in larval development was compensated resulting in a similar number of hatched flies on day 12 irrespective of the food source (for (**C**,**D**): *n* = 3 replicates with *N* = 180 eggs per condition). Data are shown as boxplots with whiskers indicating minimal and maximal values. Statistical significance was assumed at * *p* < 0.05, ** *p* < 0.01 (ANOVA). HFD-BF: high-fat diet with butterfat, HFD-SO: high-fat diet with sunflower oil, HFD-OO: high-fat diet with olive oil, HFD-LO: high-fat diet with linseed oil, HFD-FO: high-fat diet with fish oil.

**Figure 9 cells-11-04043-f009:**
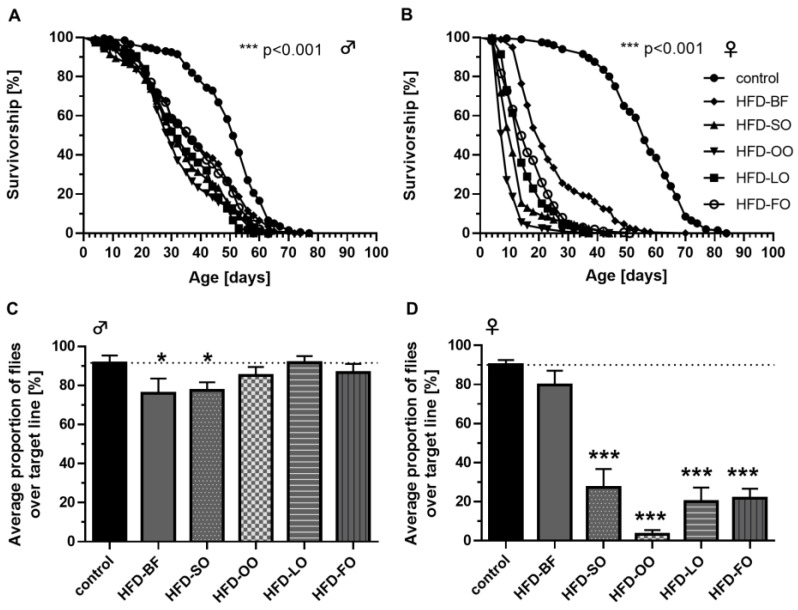
The impact of different high-fat diet (HFD) qualities on the lifespan and climbing activity of *D. melanogaster.* Lifelong ingestion of an HFD comprising either 12% (*w*/*v*) butterfat (BF), sunflower oil (SO), olive oil (OO), linseed oil (LO) or fish oil (FO) significantly reduced the lifespan of (**A**) male and (**B**) female *D. melanogaster* compared to controls. *D. melanogaster w^1118^* flies were fed ad libitum with the indicated experimental diets, while food was renewed 3 times a week. Flies were maintained in horizontally placed vials under standard conditions (12/12-h light dark cycle; 25 °C; 60% humidity) throughout lifetime. (*n* = 2; *N* = 200 flies). Significance was assumed at *** *p* < 0.001 (logrank). (**C**,**D**) *D. melanogaster* were maintained under standard conditions and fed ad libitum for 3 days on a standard medium and different experimental HFDs comprising 12% (*w*/*v*) of the indicated fat source, respectively. (**C**) The climbing activity of male flies was slightly diminished by an HFD containing 12% butterfat (BF) and sunflower oil (SO), respectively. (**D**) Ingestion of an HFD comprising 12% (*w*/*v*) SO, olive oil (OO), linseed oil (LO) or fish oil (FO) significantly reduced the inducible climbing activity of females when compared to controls. Bar represent the average proportion of flies reaching the target line (*n* = 3; *N* = 120 flies per treatment). Statistical significance was assumed at * *p* < 0.05, and *** *p* < 0.001 (ANOVA).

**Figure 10 cells-11-04043-f010:**
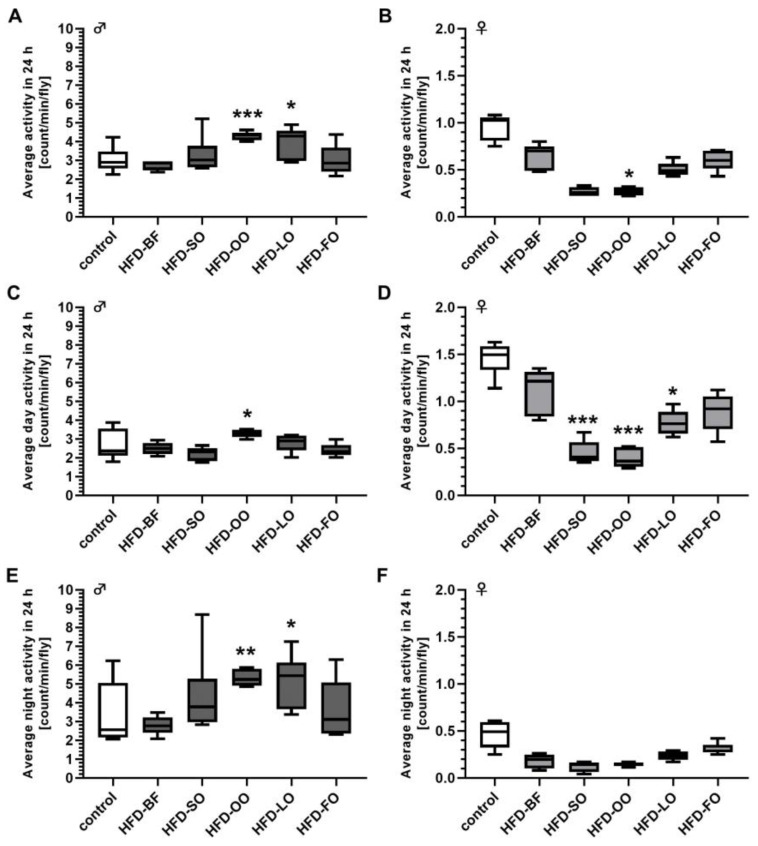
The impact of different high-fat diet (HFD) qualities on the spontaneous locomotor activity of male and female *D. melanogaster*. *D. melanogaster w^1118^* flies were fed ad libitum for 3 days with a standard sugar yeast medium (controls) or an HFD comprising 12% (*w*/*v*) butterfat (BF), sunflower oil (SO), olive oil (OO), linseed oil (LO) and fish oil (FO), respectively. Flies were maintained under standard conditions throughout the experimental treatment. The spontaneous activity was recorded with a Drosophila Activity Monitoring system at intervals of 10 min after an adaptation period of 24 h. (**A**) The average spontaneous activity per fly was significantly increased in males following intake of an HFD-OO and HFD-LO compared to the controls. (**B**) The average spontaneous activity in females was reduced following ingestion of an HFD-OO compared to their control-treated counterparts. (**C**,**E**) In male flies supplied an HFD-LO the spontaneous activity at night-time was increased, whereas in HFD-OO fed males an enhanced activity at day- and night-time was observed in comparison to controls. (**D**,**F**) The spontaneous daytime activity in females was decreased following the ingestion of an HFD-SO, HFD-OO and HFD-LO in comparison to controls, respectively. Experiments were carried out independently three times with *N* = 120 flies per treatment group. Data are shown as boxplots with whiskers indicating minimal and maximal values. Statistical significance was assumed at * *p* < 0.05, ** *p* < 0.01, and *** *p* < 0.001 (ANOVA).

**Figure 11 cells-11-04043-f011:**
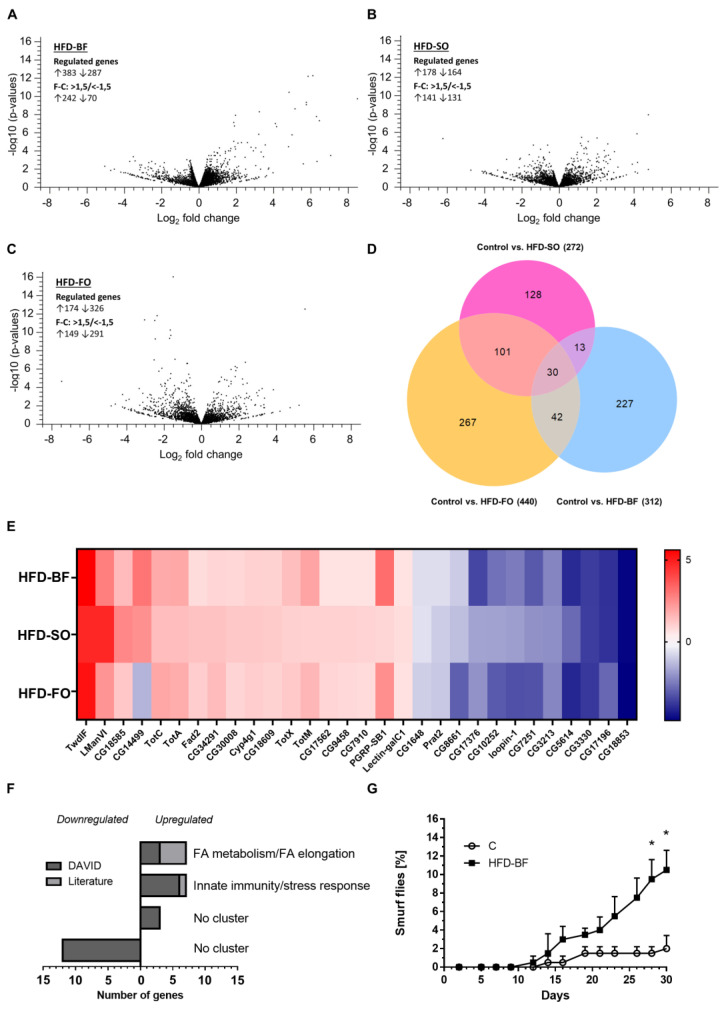
Transcriptional alterations in female *D. melanogaster* following short-term intake of a high-fat diet with either 12% (*w*/*v*) butterfat (HFD-BF), sunflower oil (HFD-SO) or fish oil (HFD-FO). Female *D. melanogaster w^1118^* were fed ad libitum with SY control diet or experimental HFDs additionally supplemented with 12% (*w*/*v*) butterfat (HFD-BF), sunflower oil (HFD-SO) or fish oil (HFD-FO). Flies were maintained under standard conditions for 3 days, before they were harvested for RNA isolation from whole-body lysates. RNAs were sequenced by a HiSeq3000/HiSeq 4000 system. (**A**–**C**) The volcano plots represent the log_2_ fold change in gene expression plotted against the log_10_ (*p*-value) for the different HFD groups in comparison to the control group. The numbers of significantly regulated genes without and with considering a minimum fold change threshold of >1.5/<−1.5 are depicted. (**A**) More genes were significantly upregulated than downregulated by the ingestion of an HFD-BF. (**B**) The transcription of a similar number of genes was significantly up- and downregulated following the dietary intake of an HFD-SO. (**C**) The ingestion of an HFD-FO mainly downregulated the expression of target genes. (**D**) In the Venn diagram, overlaps of significant alterations in gene expression after dietary intake of either an HFD-BF, HFD-SO or HFD-FO in comparison to the control group are depicted (fold change > 1.5/< −1.5). (**E**) The heat map shows the log_2_ fold change (>1.5/<−1.5) of the 30 genes that represent the intersection of all genes significantly regulated by the different HFD qualities. Thereof, the expression of 18 genes was upregulated and that of 12 genes was downregulated. (Statistical significance was assumed at * *p* < 0.05: Significant differences between groups, evaluated by Wald test). (**F**) Using DAVID bioinformatics software in combination with a subsequent flybase and literature search, an enrichment analysis of the 30 genes revealed the 2 functional clusters “FA metabolism/FA elongation” and “innate immunity/stress response” among the genes that were upregulated in female fruit flies by all 3 fat qualities. (**G**) The impact of feeding an HFD-BF on the gut integrity of female fruit flies was investigated over time by using the smurf assay. The cumulative increase in the percentage of animals that exhibit a smurf phenotype is depicted (*n* = 2 independent experiments with *N* = 200 animals per treatment group). Statistical significance was assumed at * *p* < 0.05 (Multiple *t*-test).

**Table 1 cells-11-04043-t001:** The impact of increasing concentrations of dietary butterfat on the lifespan data for male and female fruit flies. The median, mean and maximum lifespan data for male and female *w^1118^* fruit flies fed a control sugar yeast (SY) diet are listed as the mean values ± SEM in days (d) (for each sex *n* = 3; *N* = 450 animals). The percentage differences from these values in response to increasing BF concentrations are shown.

		Males			Females	
	Median	Mean	Maximum	Median	Mean	Maximum
**Control**	55.0 ± 2.1 d	54.0 ± 2.0 d	85.3 ± 4.6 d	62.7 ± 3.4 d	60.7 ± 4.3 d	97.0 ± 3.1 d
**HFD-3**	−5.5%	−4.9%	−2.3%	−5.9%	−5.5%	−6.5%
**HFD-6**	−5.5%	−5.6%	−4.7%	−8.5%	−10.4%	−8.9%
**HFD-9**	−10.3%	−8.6%	−4.7%	−16.0%	−14.3%	−13.1%
**HFD-12**	−10.9%	−12.3%	−13.3%	−22.9%	−19.2%	−10.7%

HFD-3: high-fat diet with 3% (*w*/*v*) BF, HFD-6: high-fat diet with 6% (*w*/*v*) BF, HFD-9: high-fat diet with 9% (*w*/*v*) BF, HFD-12: high-fat diet with 12% (*w*/*v*) BF.

**Table 2 cells-11-04043-t002:** The impact of different dietary fat qualities on the lifespan of male and female fruit flies. The median, mean and maximum lifespan data obtained for male and female *w^1118^* fruit flies fed a control sugar yeast (SY) diet are listed as the mean values in days (for each sex *n* = 2; *N* = 200 animals). The percentage differences from these values in response to the supplemented fat quality are shown. HFD-BF: high-fat diet with butterfat, HFD-SO: high-fat diet with sunflower oil, HFD-OO: high-fat diet with olive oil, HFD-LO: high-fat diet with linseed oil, HFD-FO: high-fat diet with fish oil.

	Males			Females		
	Median	Mean	Maximum	Median	Mean	Maximum
**Control**	52.0	50.0	75.5	56.0	55.5	82.5
**HFD-BF**	−29%	−24%	−9%	−63%	−56%	−27%
**HFD-SO**	−38%	−34%	−20%	−82%	−77%	−48%
**HFD-OO**	−44%	−36%	−17%	−84%	−82%	−54%
**HFD-LO**	−39%	−32%	−25%	−74%	−72%	−56%
**HFD-FO**	−29%	−26%	−17%	−71%	−69%	−39%

**Table 3 cells-11-04043-t003:** Enrichment analysis of the transcriptional changes induced in *D. melanogaster* females by an HFD-BF, HFD-FO and HFD-SO, respectively. Female *D. melanogaster w^1118^* were fed ad libitum with SY control diet or experimental HFDs additionally supplemented with 12% (*w*/*v*) butterfat (HFD-BF), sunflower oil (HFD-SO) or fish oil (HFD-FO). Flies were maintained under standard conditions for 3 days before they were harvested for RNA isolation from whole-body lysates. RNAs were sequenced by a HiSeq3000/HiSeq 4000 system. The genes that were significantly up- or downregulated (threshold FDR ≤ 0.1 and fold change > 1.5/< −1.5) by the respective HDF diet were separately analyzed for functional annotation clusters using the DAVID software tool. The functional clusters obtained were populated with matching genes from a literature search. Moreover, if at least two genes identified via eye inspection could be associated with an obviously relevant biological function; they were added as orphan categories recognizable by the lack of statistical data.

**HFD-BF vs. Control** (FDR ≤ 0.1)			
**Functional cluster/** **Orphan categories**	**Number of Genes**	***p*-Value**	**Benjamini**
	*Upregulated (39)*		
Innate immunity/stress response	17 (+11 *)	1.5 × 10^−24^	4.6 × 10^−24^
FA metabolism/FA elongation	2	n.d.	n.d.
	*Downregulated (0)*		
(no cluster identified)	-	-	-
**HFD-FO vs. Control **(FDR ≤ 0.1)			
**Functional Cluster/** **Orphan categories**	**Number of Genes**	***p*-Value**	**Benjamini**
	*Upregulated (23)*		
Innate immunity/stress response	5 (+1 *)	3.7 × 10^−7^	7.3 × 10^−7^
Monooxygenase activity	3	6.6 × 10^−3^	9.5 × 10^−2^
FA metabolism/FA elongation	0 (+5 *)	n.d.	n.d.
Alkaline phosphatase activity	1 (+2 *)	n.d.	n.d.
	*Downregulated (43)*		
Proteolysis	11	3.0 × 10^−8^	8.0 × 10^−7^
Lipase	3 (+1 *)	4.6 × 10^−3^	2.5 × 10^−2^
Odorant binding	3	6.8 × 10^−3^	3.2 × 10^−2^
**HFD-SO vs. Control **(FDR ≤ 0.1)			
**Functional Cluster/** **Orphan categories**	**Number of Genes**	***p*-Value**	**Benjamini**
	*Upregulated (11)*		
FA metabolism/FA elongation	1 (+3 *)	n.d.	n.d.
	*Downregulated (4)*		
(no cluster identified)	-	-	-

* number of matching genes added due to flybase and literature search; n.d. = not determined.

## Data Availability

All data generated or analyzed during this study are included in the manuscript and Appendix A. The raw data and materials used to support the findings of this study are available from the corresponding author upon request.

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
