# Peer review of "Fat Quality Impacts the Effect of a High-Fat Diet on the Fatty Acid Profile, Life History Traits and Gene Expression in Drosophila melanogaster"

_cells, 2022, doi:10.3390/cells11244043_

Round 1

Reviewer 1 Report

I thought the work was, methodologically, quite thorough and respectable. I have 1 moderate and 1 minor criticism.

(1 - moderate) Many of the figures are very difficult to see, due to very poor image quality. I'm not sure if the authors never submitted adequate-resolution tiff/png files that will be used in the event of publication, or if the current embedding of the figures just poorly represents the resolution of the supplied images. Regardless, the really poor figure resolution cheapens an otherwise quality paper, and I recommend the authors remedy this prior to any publication. 

(2 - minor) I appreciated the authors' empirical calculation of energy intakes from the flies' varying diets. I thought this was a rigorous way to account for calorie-dependent and -independent effects of the diets. Alas, I felt the nuance of these contributions weren't discussed by the authors as much as they could have been. Despite this, I consider it a minor criticism, since the authors don't over-interpret the results' caloric dependence vs independence. 

Author Response

Dear reviewer,

We are grateful to the very helpful comments, which enabled us to improve the quality of our paper. You will find our response to your comments below.

Responses to Reviewer 1

Comments and Suggestions for Authors

I thought the work was, methodologically, quite thorough and respectable. I have 1 moderate and 1 minor criticism.

1 – moderate point. Many of the figures are very difficult to see, due to very poor image quality. I'm not sure if the authors never submitted adequate-resolution tiff/png files that will be used in the event of publication, or if the current embedding of the figures just poorly represents the resolution of the supplied images. Regardless, the really poor figure resolution cheapens an otherwise quality paper, and I recommend the authors remedy this prior to any publication. 

We apologize for the poor resolution of the uploaded figure files. We have now uploaded figure files (.png) of higher quality.

2 - minor point. I appreciated the authors' empirical calculation of energy intakes from the flies' varying diets. I thought this was a rigorous way to account for calorie-dependent and -independent effects of the diets. Alas, I felt the nuance of these contributions weren't discussed by the authors as much as they could have been. Despite this, I consider it a minor criticism, since the authors don't over-interpret the results' caloric dependence vs independence. 

We agree with the reviewer’s opinion that the energy intake data should not be over-interpreted, especially in the light of the oily/fatty fecal spots, which most probably indicate a low rate of procession of the ingested fat. Hence, we have not added any further speculations on calorie-dependent and calorie-independent effects of a HFD to the manuscript.

Reviewer 2 Report

The authors aimed to investigate the effects of fat content and fat quality on lifespan, climbing activity, fertility, fatty acid profile, and gene expression of flies. The found that body weight and body composition were not altered by BF concentration ranging from 3% to 12%, while health parameters such as lifespan, fecundity and larval development were impaired in a dose-dependent manner. The also found that all 12% HFDs diets (BF, sunflower oil, olive oil, linseed oil, fish oil) shortened lifespan of flies. Moreover, fatty acid profiles and differentially expressed genes varied according to the dietary fat qualities. 

Comments and suggestions for authors:

1. Major points

1.1. The author tested different dietary fat ranging from high PUFA (HFD-SO, HFD-LO, HFD-FO) over high MUFA (HFD-OO) to high SFA (HFD-BF), and found that they persistently impaired lifespan and some healthspan related measures. The results were inconsistent with previous studies that PUFA and MUFA were conducive to health and longevity [PMID: 28379943, PubMed: 23392608, PubMed: 21308420]. Please discuss these differences.

1.2 The author stated that “we observed fatty/oily fecal spots when fruit flies were exposed to HFD which indicates that most of the ingested fat is excreted unprocessed”, if so, why not test a wider range of dietary fat content based on reports that energy from high fat diet in fly was commonly 15-30%?

2. Minor points

1.1 Over a half of cited references did not come from recent publications (within the last 5 years), the authors may be suggested to update some cited references.

1.2 Please include details regarding the detection method of fatty acid profile in the Materials and Methods section.

1.3 The author stated that The data evaluation was initiated with the definition of appropriate statistical mixed models for the measurement variables” in the Method section, however, the mixed model was not used in all data analysis.

1.4 Figure 1C, HFD-12 did not impact the body weight in female D. melanogaster, however in Figure 7F, FD-BF robustly decreasead the body weight of female flies, please clarify the controversial result.

Author Response

Dear reviewer,

We are grateful to the very helpful comments, which enabled us to improve the quality of our paper. You will find our response to your comments below.

Responses to Reviewer 2

Comments and Suggestions for Authors

The authors aimed to investigate the effects of fat content and fat quality on lifespan, climbing activity, fertility, fatty acid profile, and gene expression of flies. The found that body weight and body composition were not altered by BF concentration ranging from 3% to 12%, while health parameters such as lifespan, fecundity and larval development were impaired in a dose-dependent manner. The also found that all 12% HFDs diets (BF, sunflower oil, olive oil, linseed oil, fish oil) shortened lifespan of flies. Moreover, fatty acid profiles and differentially expressed genes varied according to the dietary fat qualities. 

Comments and suggestions for authors:

  1. Major points

Major point 1.1. The author tested different dietary fat ranging from high PUFA (HFD-SO, HFD-LO, HFD-FO) over high MUFA (HFD-OO) to high SFA (HFD-BF), and found that they persistently impaired lifespan and some healthspan related measures. The results were inconsistent with previous studies that PUFA and MUFA were conducive to health and longevity [PMID: 28379943, PubMed: 23392608, PubMed: 21308420]. Please discuss these differences.

We thank the reviewer for raising this important point. We have addressed this issue by including a paragraph on the lifespan extending and health promoting effects of certain FA. However, we have to emphasise that in C. elegans the effect of MUFA and PUFA supplementation was not studied under HFD conditions, since FA were supplemented at concentrations far below 1% (0.8 mM oleic acid (0.02%), 10 µM arachidonic acid (0.0003%)).

Line 886-896

Epidemiological studies and human clinical trials indicate that a diet rich in MUFAs such as the Mediterranean diet with a high OO consumption rate decreases the risk of metabolic syndrome and cardiovascular disease and, hence, promotes health span [48]. In line with this, when tested under non-HFD conditions, individual MUFAs (oleic acid, palmitoleic acid or cis-vaccenic acid) supplemented at a concentration of 0.8 mM (< 0.05%; w/v) were found to prolong the lifespan of the model organism Caenorhabditis elegans [49]. Similarly, lifespan extensions were obtained when 10 µM ω-6 PUFAs (arachidonic acid; di-homo-γ-linoleic acid) (< 0.0005%; w/v) were individually added to the nematode medium [50]. Therefore, it would be interesting to perform similar non-HFD supplementation studies in D. melanogaster using a chemically defined medium [17].

Major point 1.2 The author stated that “we observed fatty/oily fecal spots when fruit flies were exposed to HFD which indicates that most of the ingested fat is excreted unprocessed”, if so, why not test a wider range of dietary fat content based on reports that energy from high fat diet in fly was commonly 15-30%?

We have chosen an upper supplementation limit of 12% fat/oil, because especially in the cases of sunflower oil, linseed oil and fish oil increasing the fat content resulted in very oily liquid consistencies of the food, which were not compatible with fruit fly maintenance. Moreover, our data demonstrate that for all fat qualities a concentration of 12% in the diet was sufficient to affect life-history traits such as lifespan and induced locomotor activity.

  1. Minor points

Minor point 1.1 Over a half of cited references did not come from recent publications (within the last 5 years), the authors may be suggested to update some cited references.

According to the reviewer’s suggestions, we have replaced the following references by more recent publications:

Drewnowski, A., Nutrition transition and global dietary trends. Nutrition 2000, 16, 486–487.

was replaced by

Kopp, W., How western diet and lifestyle drive the pandemic of obesity and civilization diseases. Diabetes, metabolic syndrome and obesity : targets and therapy 2019, 12, 2221-2236.

FAO, Fats and fatty acids in human nutrition.: Report of an expert consultation. FAO Food Nutr. Pap. 2010, 91, 1–166.

and

Pitts, M.; Dorling, D.; Pattie, C., Oil for food: The global story of edible lipids. Journal of World-Systems Research 2007, 12–32.

were replaced by

Meijaard, E.; Abrams, J.F.; Slavin, J.L.; Sheil, D., Dietary fats, human nutrition and the environment: Balance and sustainability. Frontiers in nutrition 2022, 9, 878644.

and

Gillingham, L.G.; Harris-Janz, S.; Jones, P.J., Dietary monounsaturated fatty acids are protective against metabolic syndrome and cardiovascular disease risk factors. Lipids 2011, 46, 209-228.

Warwick, Z.S.; Schiffman, S.S., Role of dietary fat in calorie intake and weight gain. Neuroscience & Biobehavioral Reviews 1992, 16, 585–596.

was replaced by

Warwick, Z.S., Dietary fat dose dependently increases spontaneous caloric intake in rat. Obesity research 2003, 11, 859-864.

In this context, we have slightly modified the text in line 39-43:

…Accordingly, global dietary recommendations for humans consider a reduction of total fat (< 30% of energy), saturated fat (< 10% of energy) and trans-fat intake (< 1% of energy) by replacement with unsaturated fats for a healthy diet [6,7]. The exchange of saturated fatty acids (SFAs) with mono- and polyunsaturated fatty acids (MUFAs, PUFAs) is usually recommended to reduce chronic diseases related to the so-called metabolic syndrome. However, in particular, the impact of total fat and saturated fat in the diet has been the object of debate [4] and…

Minor point 1.2 Please include details regarding the detection method of fatty acid profile in the Materials and Methods section.

According to the reviewer’s suggestions, we have extended the paragraph on GC-FID analyses as follows:

Line 148-162

…FAME were analyzed by capillary gas chromatography (GC) using a Shimadzu GC-2010 Plus GC system equipped with a split/splitless injector AOC-20i, an auto-mated liquid sampler AOC-20s and a flame ionization detector (Shimatsu GmbH, Duisburg, Germany). The inlet was operated in constant flow mode with helium carrier gas and a 100:1 split ratio. 1 µl of each sample was separated on a capillary GC column (Zebron® ZB-FAME, 0,20 µm, 30 m x 0,25 mm ID; Phenomenex Ltd, Aschaf-fenburg, Germany) at a flow rate of 1.2 ml/min with a total run time of 30.33 min employing a temperature program starting with an initial temperature of 80°C for 2 min, heated by a rate of 15°C/min to 140°C, then elevated by a heat rate of 2.5°C/min to 190°C, and finally heated by 30°C/min to 260°C, before the temperature was hold for 2 min). FAMEs were analyzed with the flame ionization detector (detector temperature 270 °C) according to ASU L 13.00-45/46:2018-06 and DIN EN ISO 12966-1/4:2015-11 [SGS Analytics Germany GmbH, Jena]. Peak identification was carried out via external standards (Merck-Supelco, Darmstadt, Germany). Data were calculated as % identified FAME…

Minor point 1.3 The author stated that “The data evaluation was initiated with the definition of appropriate statistical mixed models for the measurement variables” in the Method section, however, the mixed model was not used in all data analysis.

The reviewer is right. However, we have already listed all assays in the text where the mixed model was applied:

Line 284-287

…The data evaluation was initiated with the definition of appropriate statistical mixed models [43] for the measurement variables food intake, energy intake, body weight, TAG, protein, glucose content, climbing success probability, average activity, number of eggs, oviposition preference index, development time and number of pupae/hatched flies from eggs…

 The static methods for the remaining assays are also already mentioned in the corresponding paragraph 2.15.

Minor point 1.4 Figure 1C, HFD-12 did not impact the body weight in female D. melanogaster, however in Figure 7F, FD-BF robustly decreasead the body weight of female flies, please clarify the controversial result.

We thank the reviewer for bringing up this issue. The body weight of female flies fed a butterfat based 12% HFD was reduced in both cases (Figure 1C: BW controls = 1.18 mg/female fly vs BW HFD-12 = 1.09 mg/female fly; Figure 7F: BW controls: 1.21 mg/female fly vs BW HFD-12 = 1.07 mg/female fly), however, did not reach statistical significance in Figure 1C.

Accordingly, we have changed the title of paragraph 3.1 in the Result section and the statement on the impact of HFD on the body weight as follows:

Line 334/335

3.1. Increasing Concentrations of Dietary Butterfat Significantly Increased the Energy Intake without Affecting the Body Composition of Male and Female D. melanogaster

Line 347-353

The body weight of male flies was not affected by any HFD-BF feeding (Figure 1C), while female flies exhibited an even reduced body weight at higher HFD-BF concentrations (controls = 1.18 mg/female fly vs. BW HFD-12 = 1.09 mg/female fly; Figure 1F), however, without reaching statistical significance. Moreover, irrespective of the high energy intake, neither male (Figure 1G-I) nor female flies (Figure 1J-L) that received the HFDs (HFD-3 to HFD-12) differed significantly in terms of their body composition.

Round 2

Reviewer 2 Report

Minor points

Minor point 1.1. Please include the nutritional composition of different dietary culture medium.

Minor point 1.2. Athough the author have listed all assays in the text where the mixed model was applied. However, the mixed model was not found in all figure analyses. In general, the mixed model was applied to the population study due to the cofounders, but not the animal study for the precise control in experimental environment and intervention conditions. So please cite some animal study using this model, if not, please do not use it.

Author Response

Dear Editors,

Please find below and attached as a pdf-file our reply to the review report of reviewer 2 concerning our manuscript entitled

“Fat quality impacts the effect of a high-fat diet on the fatty acid profile, life history traits and gene expression in Drosophila melanogaster”

by Virginia Eickelberg, Gerald Rimbach, Yvonne Seidler, Mario Hasler, Stefanie Staats, and Kai Lüersen

which is being resubmitted to “Cells” as a contribution for the special issue “Cellular and Molecular Control of Lipid Metabolism”.

We are grateful for the reviewer’s comments and hope that the revised version of our manuscript is now suitable for publication in “Cells”.

On behalf of all authors, yours sincerely,

Kai Lüersen

Minor point 1.1. Please include the nutritional composition of different dietary culture medium.

According to the reviewer’s suggestion, we have specified the nutritional composition of the culture media used in our studies. According to the product information published on https://www.kisker-biotech.com/article/789196, the yeast extract contained per 100 g: 52 g protein, 33 g complex carbohydrates, 5 g fat, of which 1-2 g are saturated and 2-4 g monounsaturated, and various vitamins in the mg range as well as 6 g ash (minerals) and 4 g moisture. Hence, we revised the text on page 3, line 104-112 and line 127-131:

…Flies were maintained on modified Caltech (CT) medium (6.0% dextrose, 3.0% sucrose [Carl Roth, Karlsruhe, Germany], 6.0% cornmeal [Kisker, Steinfurt, Germany], 3.0% inactive dry yeast (contributing 1.6% protein, 1% complex carbohydrates and 0.15% fat as well as vitamins and minerals according to the product information), 1.0% agar, 1.0% nipagin [Genesee Scientific, San Diego, USA] (dissolved in ethanol (20% w/v) [VWR, Radnor, USA]) and 0.3% propionic acid [Carl Roth, Karlsruhe, Germany]) in Drosophila storage bottles [Kisker, Steinfurt, Germany] at 25°C and 60% relative humidity with a 12/12 h light/dark cycle in a climate chamber [Memmert, Schwabach, Germany] [32]…

…The different experimental HFDs were based on a standard sugar yeast (SY) medium (control medium) containing 10% sucrose, 10% inactive dry yeast (contributing 5.2% protein, 3.3% complex carbohydrates and 0.5 % fat, of which 0.1 - 0.2 % are SFAs and 0.2 - 0.4 % MUFAs, as well as vitamins and minerals; product information, Genesee Scientific), 2.0% agar, 1.5% nipagin (dissolved in ethanol (20% w/v)) and 0.3% propionic acid…

Minor point 1.2. Athough the author have listed all assays in the text where the mixed model was applied. However, the mixed model was not found in all figure analyses. In general, the mixed model was applied to the population study due to the cofounders, but not the animal study for the precise control in experimental environment and intervention conditions. So please cite some animal study using this model, if not, please do not use it.

The reviewer has expressed doubts about the suitability of the mixed model used by us. If we understood him correctly, he is of the opinion that the mixed model can only be used for population studies. This is a critical point raised by the reviewer that we would like to address as follows:

All measurement variables coming from the same experiment are influenced by the same experimental design, irrespective of what they represent. The experimental design must always be taken into account by the statistical analysis, if possible. Typically, influence factors like repetition, site, block etc. are regarded as a random factor if they were present within the experiment but not of interest. Hence, the use of a mixed model follows from the experimental design. It does not follow from the measurement variable. We applied mixed models for the measurement variables: food intake, energy intake, body weight, TAG, protein, glucose content, climbing success probability, average activity, number of eggs, oviposition preference index, development time, number of pupae/hatched flies from eggs, fecundity, oviposition and development. The most of these measurement variables follow a normal distribution, some follow a binomial distribution. These distributions are also taken into account by our analysis. As we have already written, we did not use such a mixed model for the remaining assays such as e.g. the comparison of the survival curves.
